EMBO
reports

# Apicomplexan F-actin is required for efficient nuclear entry during host cell invasion

Mario Del Rosario[1] (ID), Javier Periz[1], Georgios Pavlou[2], Oliver Lyth[3], Fernanda Latorre-Barragan[1,4] (ID), Sujaan Das[1], Gurman S Pall[1], Johannes Felix Stortz[1] (ID), Leandro Lemgruber[1], Jamie A Whitelaw[5], Jake Baum[3] (ID), Isabelle Tardieux[2] (ID) & Markus Meissner[1,6,*] (ID)

## Abstract

The obligate intracellular parasites *Toxoplasma gondii* and *Plasmodium* spp. invade host cells by injecting a protein complex into the membrane of the targeted cell that bridges the two cells through the assembly of a ring-like junction. This circular junction stretches while the parasites apply a traction force to pass through, a step that typically concurs with transient constriction of the parasite body. Here we analyse F-actin dynamics during host cell invasion. Super-resolution microscopy and real-time imaging highlighted an F-actin pool at the apex of pre-invading parasite, an F-actin ring at the junction area during invasion but also networks of perinuclear and posteriorly localised F-actin. Mutant parasites with dysfunctional acto-myosin showed significant decrease of junctional and perinuclear F-actin and are coincidently affected in nuclear passage through the junction. We propose that the F-actin machinery eases nuclear passage by stabilising the junction and pushing the nucleus through the constriction. Our analysis suggests that the junction opposes resistance to the passage of the parasite's nucleus and provides the first evidence for a dual contribution of actin-forces during host cell invasion by apicomplexan parasites.

**Keywords** actin; Apicomplexa; host cell invasion; myosin; nucleus
**Subject Categories** Cell Adhesion, Polarity & Cytoskeleton; Microbiology, Virology & Host Pathogen Interaction

## Introduction

The phylum Apicomplexa consists of more than 5,000 species, most of them obligate intracellular parasites, including important human

and veterinary pathogens, such as *Plasmodium* (malaria) or *Toxoplasma* (toxoplasmosis).

During their complex life cycles, apicomplexan parasites move through different environments to disseminate within and between hosts and to invade their host cell [1]. Therefore, the invasive stages, called zoites, evolved a unique invasion device, consisting of unique secretory organelles and the parasites' acto-myosin system, the Glideosome, localised in the narrow space (~30 nm) between the plasma membrane and the inner membrane complex (IMC) [2]. Zoites actively enter the host cell by establishing a tight junctional ring (TJ) at the point of contact between the two cells. The TJ is assembled by the sequential secretion of unique secretory organelles (micronemes and rhoptries), leading to the insertion of rhoptry neck proteins (RONs) into the host cell plasma membrane (PM) and underneath [3]. On the extracellular side, the exposed domain of the RON2 member binds the micronemal transmembrane protein AMA1 exposed on the parasite surface, resulting in the formation of a stable, junctional complex [3]. The TJ is further anchored to the host cell cortex by *de novo* formation of F-actin through the recruitment of actin-nucleating proteins [4,5]. During host cell invasion, the parasites use their acto-myosin motor to pass through the TJ. However, the exact role and orientation of the parasite's acto-myosin system is still under debate [6] and intriguingly, mutants for key component of this system show residual motile and invasive capacities [7–9], the latter reflecting in large part an alternative and host cell actin-dependant mode of entry [10]. According to the Glideosome model, the force generated for motility and invasion relies exclusively on F-actin polymerised at the apical tip of the parasite by the action of Formin-1 and translocated within the narrow space (~30 nm) between the IMC and PM of the parasite [11]. However, recent studies suggest that the parasite can also use other motility systems, such as a secretory-endocytic cycle that produces retrograde membrane flow [12], similar to the fountain flow model suggested for other eukaryotes [13,14].

In support of the linear motor model, was the detection of parasite F-actin underneath the junction formed by invading parasites

1 Wellcome Centre For Integrative Parasitology, Institute of Infection, Immunity & Inflammation, Glasgow Biomedical Research Centre, University of Glasgow, Glasgow, UK
2 Institute for Advanced Biosciences, CNRS, UMR5309, INSERM U1209, Université Grenoble Alpes, Grenoble, France
3 Department of Life Sciences, Imperial College London, London, UK
4 Faculty of Science, Food Engineering and Biotechnology, Technical University of Ambato, Ambato, Ecuador
5 CRUK Beatson Institute, Glasgow, UK
6 Experimental Parasitology, Department for Veterinary Sciences, Ludwig-Maximilians-University Munich, Munich, Germany
*Corresponding author. Tel: +49 89 21803622; E-mail: markus.meissner@lmu.de

when using an antibody preferentially recognising apicomplexan F-actin. Furthermore, the detection of cytosolic locations, predominantly around the nucleus [15], suggests additional roles of this cytoskeletal protein during invasion.

While it was assumed a major role of F-actin in driving Apicomplexa zoite gliding motility and cell invasion, recent studies demonstrated the pivotal role of F-actin in multiple other processes, such as apicoplast inheritance [16], dense granule motility [17] and likely nuclear functions through the control of expression of virulence genes in malaria parasites [18]. However, building a comprehensive model for F-actin dynamics, localisation and function in apicomplexan parasites has been hampered for decades by the lack of tools enabling reliable F-actin detection. Interestingly, several studies suggested that F-actin is interacting with subpellicular microtubules [19] and/or the subpellicular matrix of the IMC [20,21]. Furthermore, components of the Glideosome, such as GAPM proteins [22] or the invasion-critical myosin, MyoH were demonstrated to interact with microtubules [23], suggesting a coordinated action of the actin and microtubule cytoskeleton during host cell invasion.

With the adaptation of nanobodies specifically recognising F-actin, it is now possible to analyse F-actin dynamics in apicomplexan parasites [24,25] leading to the identification of distinct cytosolic networks of dynamic actin in both *T. gondii* and *P. falciparum* [11,24,25]. We originally identified two F-actin polymerisation centres from where most of the F-actin flow occurs in intracellular parasites [24], later expanded to three based on further studies [11,25]. One polymerisation centre can be found at the apical tip, corresponding well to the described location of Formin-1, and one close to the parasite's Golgi complex, corresponding to the location of Formin-2 [11,25]. In *T. gondii,* these three polymerisation centres of F-actin correlate well with the location of Formins-1, 2 and 3 in *T. gondii* [11]. Formins-2 and 3 appear to have overlapping functions during the intracellular development of the parasite for material exchange and formation of an intravacuolar network in intracellular parasites, while Formin-1 (FH-1) has been explicitly implicated in the apical polymerisation of F-actin required for gliding motility and cell invasion [11]. In this study, although not directly shown, the authors suggested that FH-1 polymerises F-actin at the apical tip of the parasite that is exclusively transported by the action of the Glideosome within the subalveolar space to the posterior pole. Interestingly, at least in intracellular parasites, the majority of F-actin dynamics appears to occur within the cytosol of the parasite close to the parasite's Golgi, forming a continuous dynamic flow from centre to the periphery of the parasite and from the apical to the basal pole [24,25]. In addition, measurement of peripheral F-actin flow demonstrated that it can occur bidirectional from the apical to the posterior pole and vice versa [25].

In good agreement with a cytosolic oriented F-actin system, previous reports demonstrated parasite F-actin to be associated with the subpellicular microtubules that are connected to and stabilise the IMC [19–21,26]. Furthermore, gliding-associated proteins identified via co-immunoprecipitation with the Glideosome were recently found to play important roles in stabilising the IMC and directly connecting it to the subpellicular microtubules [22,27], suggesting that a close connection exists between parasite F-actin, the Glideosome and the subpellicular network.

Here we compared F-actin dynamics in WT and mutant parasites for the acto-myosin system and correlated its dynamic location with the phenotypic consequences during host cell invasion. Interestingly, in unstimulated parasites, F-actin flow in extracellular parasites appears to be bidirectional, with the majority of events occurring in a retrograde direction, which is similar to measurements performed on intracellular parasites [25]. The activation of calcium signalling results in a shift towards retrograde F-actin flow, resulting in F-actin accumulation at the posterior pole. Comparison of F-actin dynamics in parasite mutants for the acto-myosin system helped to identify the nucleus as a major obstacle for efficient host cell invasion that needs to be squeezed through the junction using F-actin dynamics to translocate and potentially deform the nucleus through this process.

Our analysis strongly suggests a push-and-pull mechanism for nuclear entry during host cell invasion, in analogy to the dynamics observed during migration of other eukaryotes through a constricted environment [28].

# Results

### Rational for selecting parasite mutants in order to analyse F-actin dynamics during host cell invasion

Over recent years, conditional mutagenesis systems have been employed to generate and characterise a whole assortment of parasite mutants affected in host cell invasion. These mutants can be affected in different, unrelated pathways, such as microneme secretion, formation of the tight junction or host cell entry. Since we were interested in analysing F-actin dynamics during the invasion process, we chose parasite mutants that were previously characterised to be significantly affected, but not blocked in host cell entry and linked to the parasites acto-myosin system. We excluded all mutants that have been described to be completely blocked in the initiation of invasion, such as MyoH [23] or the micronemal protein MIC8 [29], since no invasion event can be expected that can be analysed. Similarly, we also excluded mutants involved in actin dynamics that have not been implicated in host cell invasion, such as FH-2,3 [11] or myosins not involved in host cell invasion, such as MyoI [30].

Therefore, we chose parasite actin [7,9], the actin depolymerisation factor ADF [31] and myosin A (MyoA), which has been used previously to successfully identify the contribution of the host cell for parasite invasion [10].

### F-actin flow is mainly organised within the cytosol of extracellular parasites and is modulated by Calcium and cGMP signalling

We previously imaged F-actin flow in intracellular parasites and were able to discriminate two major polymerisation centres, one at the apical tip and one close to the apicoplast, where F-actin is formed and subsequently transported to the posterior end of the parasite [24]. Indeed, recent studies on *Toxoplasma* and *Plasmodium* demonstrated that Formin-2 is localised to the apicoplast of the parasite and required for most of the intracellular F-actin dynamic [11,25], while Formin-1 is localised at the apical tip, where it is thought to be exclusively required for parasite motility and invasion. In good agreement, in intracellular parasites, F-actin appears

to be formed at two polymerisation centres, localised at the apical tip and close to the Golgi region of the parasite, indicating that FH-1 and FH-2 are acting as nucleators during intracellular parasite development. Intriguingly, the majority of F-actin dynamics in intracellular parasites occurs within the cytosol of the parasite with F-actin flow occurring in a bidirectional manner (retrograde and anterograde) [24,25]. We were interested to determine if during the transient extracellular life and at the time of cell invasion, the patterns of actin dynamics differ from the intracellular ones and how they were impacted upon disruption of the Glideosome or factors critically involved in F-actin regulation.

Since transient transfections are typically associated with overexpression of the target protein hence with significant uncontrolled impact on a dynamic equilibrium [24], we generated *T. gondii* lines stably expressing Cb-EmeraldFP. We ensured that the expression of the chromobody was comparable between the different parasite lines and ensured that its expression does not lead to phenotypic alterations and misleading analysis. We established transgenic parasites expressing Cb-EmeraldFP in WT parasites, a null mutant for myosin A (MyoA), the core motor of the Glideosome [16] and a conditional mutant for the critical actin regulator, actin depolymerisation factor ADF, *adf*cKD [31]. Using live 3D-structured illumination microscopy (3D-SIM), we compared F-actin dynamics and found two discernible F-actin polymerisation centres (Fig 1A) as previously seen in intracellular parasites [24]. Time-lapse analysis demonstrated that the majority of F-actin dynamics occurs in the cytosol of the parasite with some F-actin flow detectable at the periphery occurring in retrograde and anterograde orientation, similar to intracellular parasites (see also [25]). In good agreement with the localisation of FH-1 and FH-2, two polymerisation centres can be detected. F-actin flow starts from the apical tip (1$^{st}$ polymerisation centre, yellow arrow in Fig 1A) to the second polymerisation centre close to the Golgi (red arrow, Fig 1A) to the posterior pole of the parasite (Fig 1A, Movie EV1). While disruption of *myoA* did not result in significant changes of F-actin localisation or dynamics in resting parasites (Fig 1A, Movie EV1), depletion of ADF completely abrogated actin dynamics, F-actin accumulation being observed at both apical and posterior parasite poles (Fig 1A, Movie EV1).

To map the actin structures visualised with the chromobody, we employed skeletonisation processing [32], which converts F-actin signals into individual pixels, that can be traced to obtain information about the localisation and dynamics of F-actin. The resulting skeletonised movie can be projected to show where the majority of (detectable) F-actin dynamics occur during the timeline of the movie (Fig 1B, Movie EV1). This analysis confirmed that parasite F-actin formed at the apical tip (yellow arrowhead) and Golgi region (red arrowhead), converge within the cytosol of the parasite and reach the basal pole of the parasite. Interestingly, both peripherical and cytoplasmic flow appear to be connected and do not occur independently from each other (Fig 1B). Note that both WT and *myoA*KO parasites displayed similar signals, whereas the ADF—conditional mutant did not.

Next, kymograph analysis was performed to analyse individual F-actin flow events using KymographClear and KymographDirect [33] confirming the presence of a continuous dynamic F-actin network in cytosolic location that connects the apical pole, Golgi area and basal pole of the parasite, in good agreement with the location of the 3 Formins previously described in *T. gondii* [11].

KymographClear is a FIJI plugin that generates a 3 colour-coded kymograph, where each colour labels either forward movement (red), backward movement (green) or static (blue) as shown (Fig 1C and Appendix Fig S1). This kymograph can then be read into the KymographDirect software, which uses automatic detection to trace trajectories of moving particles in the image. Values are then assigned to these trajectories based on both intensity and speed [33]. The intensity can be assessed in specified tracks over time as the average intensity over time shown in Fig 1C. Both the RH and *myoA* KO presented dynamic signals connecting the apical pole (close to the point 0), Golgi region (between 2 and 3 μm) and basal pole. In the case of the *adf* KD, only static signals could be detected at the apical and basal pole of the parasite, confirming the lack of F-actin dynamics in these mutants.

Next, we used this analysis to investigate intensity profiles of actin signal travelling in the periphery of the parasite (Appendix Fig S1B–D). Interestingly, actin dynamics along the periphery is similar between WT and *myoA* KO with bidirectional actin flow events as revealed by the kymographs, while confirming previous results with the *adf* KD mutants, which shows a lack of F-actin dynamics across the entire parasites body (first half (−), Appendix Fig S1A–C).

Together, the data presented in this work point that similarly to intracellular parasites, the F-actin is highly dynamic in extracellular parasites displaying two major polymerisation centres, one at the apical tip and another at the Golgi region of the parasite. Of note, this analysis underlines that there is actin flow coming from the cytoplasm towards the periphery and shaping a highly dynamic network that connects the apical and basal pole of the parasite, therefore independently of any F-actin network positioned between the zoite's PM and IMC.

Tosetti *et al* [11] suggested that the activation of a Ca$^{2+}$-signalling cascade would lead to increased retrograde transport of F-actin, eventually accumulating posteriorly. However, we found that the treatment of parasites with the calcium ionophore A23187 induced anterior accumulation of F-actin in wild-type parasites in the majority of cases (Fig 1D–F; Appendix Fig S1A; Movie EV2), while BIPPO triggered a predominantly basal F-actin accumulation (Fig 1D–F; Appendix Fig S1A; Movie EV2). BIPPO is a drug capable of inhibiting 3′,5′-cyclic nucleotide phosphodiesterases (PDEs) responsible in blocking the breakdown of cyclic nucleotides such as cAMP and cGMP. This mechanism is believed to be responsible in the activation of microneme secretion and egress by modulating the signal pathway of PKG-dependent processes [34].

Once MyoA null mutants were submitted to treatments with A23187 or BIPPO, the F-actin remained rather evenly distributed within the cytoplasm albeit with some peripheral enrichment in the first half of the parasite closer to the apical end (Fig 1D–F; Movie EV2). In contrast, depletion of ADF led to the expected phenotype with F-actin accumulating at the apical and basal pole of the parasite, independently of A23187 or BIPPO addition (Fig 1D–F; Movie EV2). Together, these data suggest that calcium signalling regulates F-actin dynamics, potentially by stimulating the activity of Formin-1 followed by F-actin transport to the posterior pole of the parasite by the action of MyoA.

In conclusion, similar to intracellular parasites, extracellular parasites show impressive F-actin dynamics within the cytosol. In parasite mutants for ADF, this dynamic is abolished, and F-actin accumulates at the apical and basal pole, independent of triggering

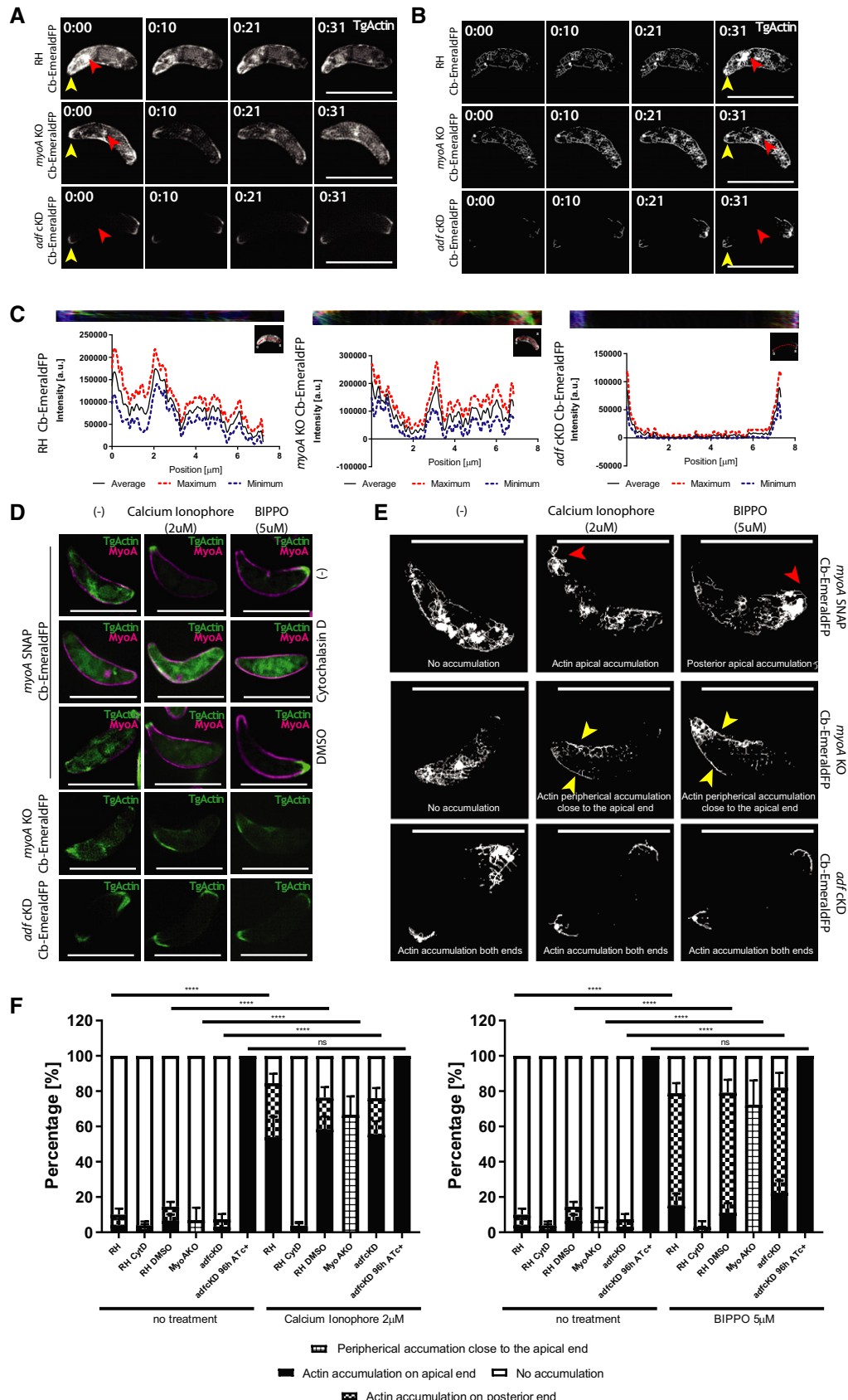

**Figure 1.**

**Figure 1.  F-actin, formed at two major nucleation centres, forms a dynamic, continuous network within the cytosol of the parasite.**

A   Stills depicting actin flow (see Movie EV1) in extracellular RH, *myoA* KO and *adf* KD parasites expressing Cb-EmeraldFP. Continuous F-actin flow can be seen from the apical tip, to the Golgi region towards the posterior pole of the parasite. No difference can be observed between RH and *myoA* KO. For *adf* KD, the flow is completely abrogated. Red arrowhead marks the Golgi region, while the yellow arrowhead marks the apical nucleation centre. The time lapse is presented in Movie EV1.

B   Skeletonisation processing of (A) depicting areas where F-actin dynamics/flow is prevalent. Individual signals for F-actin form a continuous, dynamic network that connects apical and posterior pole of the parasite. Red arrowhead marks the Golgi region nucleation centre, while the yellow arrowhead marks the apical nucleation centre. The images represent a projection of frames for each timepoint to provide a better overview of the dynamics of F-actin. The time lapse is presented in Movie EV1.

C   Kymograph analysis of (A) and (B). The kymograph was generated by tracing a line in the middle of the parasite's body (red line). The colour-coded kymograph represents forward movement (red), backward movement (green) and static F-actin (blue). The results demonstrate continuous exchange of F-actin from the apical to the posterior pole of the parasite within the cytosol of the parasite. This cytosolic exchange is similar in RH and *myoA* KO parasites, while completely abolished in the case of adf KD.

D   Top three rows: Parasites expressing Cb-EmeraldFP along with *myoA*-SNAP before and after Ca$^{2+}$ ionophore (A23187) or BIPPO treatment. The parasites were also treated with 2 µM of cytochalasin D or DMSO (as control) for 30 min. After the addition of A23187, preferential relocalisation of actin can be observed at the apical tip, while the addition of BIPPO caused F-actin accumulation at the basal end. 4$^{th}$ row: *myoA*KO parasites expressing Cb-EmeraldFP before and after treatment with A23187 and BIPPO. While no apical or basal accumulation of F-actin is observed, some peripheral location of F-actin occurs in the presence of A23187 or BIPPO, preferentially in the apical half of the parasite. Bottom row: *adf* cKD parasites expressing Cb-EmeraldFP before and after treatment with Ca$^{2+}$ ionophore or BIPPO. No apparent change in F-actin localisation can be seen upon treatment. The time lapse is presented in Movie EV2.

E   Skeletonisation of movies shown in (D). Before the addition of A23187 or BIPPO, RH and *myoA* KO parasites behave similar with no accumulation of F-actin at the apical tip. After treatment with calcium ionophore A23187, preferential accumulation of F-actin at the apical tip can be observed for RH (red arrowhead), while BIPPO presented preferential accumulation in the basal end (red arrowhead). In *myoA* KO, some relocalisation of actin is observed at the periphery of the parasite (yellow arrowhead). In the case of *adf* cKD, no relocalisation occurs. The images represent a projection of frames for each timepoint to provide a better overview of the dynamics of F-actin.

F   Quantification of actin accumulation before and after treatment with Ca$^{2+}$ ionophore or BIPPO. The parasites were counted for F-actin accumulation after adding Ca$^{2+}$ ionophore or BIPPO. Numbers were generated by counting total number of parasites and then number of parasites with actin accumulation on the apical or basal tip. A minimum of 200 parasites were counted upon 3 biological replicates. Two-way ANOVA was used for statistical analysis and Tukey's multiple comparison test. ****$P < 0.0001$. Error bars represent standard deviation.

Data information: Numbers indicate minutes:seconds, and scale bars represent 5 µm.
Source data are available online for this figure.

a calcium-signalling cascade. In contrast, disruption of MyoA results in normal F-actin dynamics, but upon calcium signalling, F-actin remains more diffusely localised within the cytosol of the parasite with some accumulation at the periphery.

**The acto-myosin system is required to ensure efficient invasion**

To investigate whether this impressive cytoplasmic F-actin dynamics could assist the parasite over the invasion of the target cell, namely when it fits through the junction and constricts (Fig 2A and B), we first measured the average diameter of parasites prior and during invasion. We observed a significant decrease at mid-invasion, (prior: 2.5–3 µm and mid-invasion 1.5–2 µm) that attests significant body deformation and compression in order to fit through the junction (Fig 2A and B). In good agreement, analysis of the average diameter of the parasite's nucleus indicated that this organelle suffers a striking deformation, up to ~50% of its normal diameter (i.e. < 1 µm) while passing through the junction (Fig 2A and B). Since the acto-myosin system generates traction force at the junction to account for the invasive force, we investigated if interference causes changes, that is differences in the diameter of the TJ and/or stronger deformations of the parasite. While a slight reduction in the average TJ diameter was observed during invasion of *myoA* KO parasites (Fig 2C), the junctional ring appears remarkably constant upon interference with the acto-myosin system of the parasite.

Previously, it was demonstrated that mutants for the acto-myosin system enter the host cell in a stop-and-go manner, leading to a prolonged invasion process with different features [7,9,31]. While mutant parasites could be driven through the TJ by the host cell membrane outward projections in an uncompleted micropinocytosis-like process [10], they could in some cases remain trapped in the TJ and die, probably due to compressive forces exerted on the

embedded part of the parasite by the host cell membrane protrusions [10]. We infer from these observations that the tachyzoite likely needs to counteract compressive forces at the TJ to secure entry of its bulky organelles, such as the nucleus and hypothesised that the parasite acto-myosin is critically involved in nuclear entry.

We compared the invasion process of WT and mutant parasites in detail using time-lapse analysis (Fig 2D and E, Appendix Fig S2, Movie EV3 and EV4). Interestingly, the invasion process of WT parasites proceeded with an initial rapid invasion step, followed by a short pause when ~1/3 of the body was inside the host cell, and another acceleration until the parasite was fully inside the host cell (Fig 2D and E, Movie EV3). In contrast, while initial invasion steps appeared to be comparable between WT parasites and mutants for the acto-myosin system (*act1*cKO, *mlc1*cKO, *myoA*KO), an extended period then followed, where parasites stalled (up to 8 min), before invasion continued (Fig 2D and E, Appendix Fig S2, Movie EV4). In some cases, we also observed abortive invasion events, where the invading parasites terminally stalled before being pushed out from the TJ (Fig 2F, Movie EV5). Interestingly, this stalling occurred approximately when the nucleus reached the TJ suggesting that in addition to providing the traction force that promotes entry through the TJ [30], F-actin also assists the passage of the nucleus through the TJ, akin to other eukaryotes when migrating through constricted environments [35].

**F-actin accumulates at the posterior pole and in most cases at the junction**

To correlate F-actin dynamics with the invasive behaviour of parasites, we stably expressed F-actin chromobodies in fusion with Emerald FP in *T. gondii* [24] WT and mutant parasites and compared the localisation and dynamics of F-actin when they

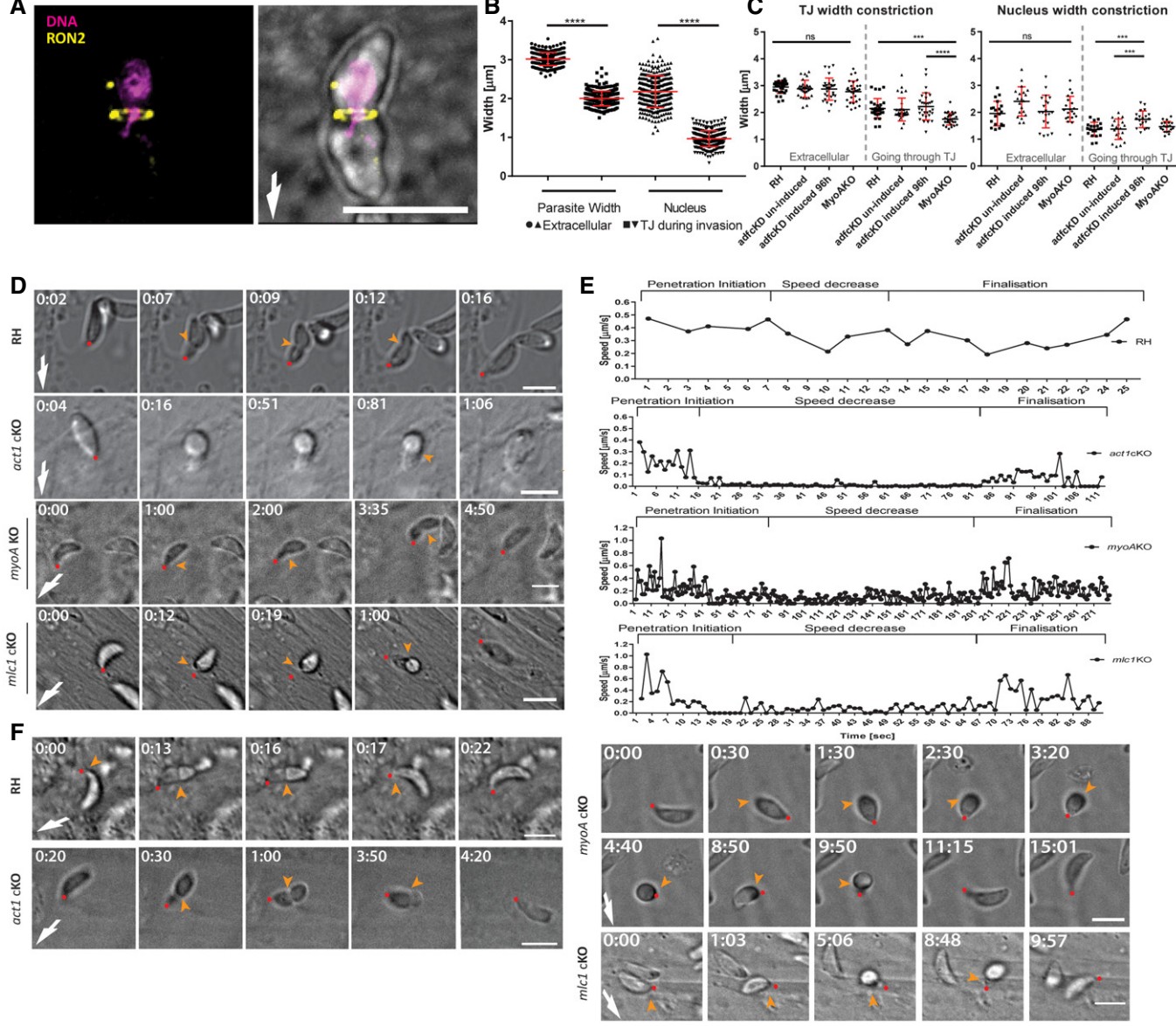

**Figure 2. The nucleus is a limiting factor for host cell invasion.**

A   Wild-type RH parasites mid-invasion. Tight junction assay was done by allowing freshly egressed parasite to invade for 5 min, fixed with 4% PFA before labelling the TJ with anti-Ron2 (yellow) and DAPI (magenta) for nuclear staining. White arrow points at invasion direction.

B   Graph showing deformation of the whole parasite and its nucleus between invading and non-invading parasites. 100 parasites found mid-invasion were counted in triplicate and compared to extracellular, freshly egressed parasites. One-way ANOVA was used for statistical analysis. ****$P < 0.0001$.

C   Graph showing deformation of the whole parasite and its nucleus between invading and non-invading parasites. Indicated parasites were analysed. For *adf* cKD, induced and non-induced conditions were compared, as indicated. 30 parasites were counted for each condition in triplicate. One-way ANOVA was used for statistical analysis and Tukey's multiple comparison test. ****$P < 0.0001$. ***$P < 0.0009$.

D   Time-lapse microscopy analysis of indicated parasites invading HFFs cells. Time was determined from the onset of invasion. The time lapse is presented in Movie EV4. White arrow points at invasion direction; yellow arrow points at tight junction; red dot follows the apical pole of the parasite.

E   Speed profiles of invading parasites shown in (D). The relative displacement of the apical tip over time allowed speed calculations. Individual stages were named in regard to the invasion step (penetration initiation, speed decrease and finalisation). Invasion was analysed with Icy Image Processing Software (Pasteur Institut) using the Manual Tracking plugin by analysing each movie 10 times and averaging the measurements (for additional speed profiles, see Appendix Fig S2).

F   Time-lapse analysis demonstrating abortive invasion of indicated parasites. Time was determined from the onset of invasion. The time lapse is presented in Movie EV5. White arrow points at invasion direction; yellow arrow points at tight junction; red dot follows the apical pole of the parasite.

Data information: Error bars represent standard deviation. Numbers indicate minutes:seconds, and scale bars represent 5 μm.
Source data are available online for this figure.

invaded the host cells. To ensure specificity of Cb-EmeraldFP for F-actin during the invasion process, we used a previously validated antibody raised against *P. falciparum* actin that preferentially recognises F-actin [15] and compared the staining obtained with this antibody on WT parasites with the staining obtained with Cb Emerald-expressing parasites. In both cases, F-actin accumulation at the junction and the posterior pole was apparent (Appendix Fig S3). Interestingly, while posterior accumulation of F-actin could always (100% of all analysed images) be detected, it was not systematic for the F-actin localised at the junction site (Fig 3A, E and F). We also ensured that the expression of Cb Emerald did not cause any significant differences in the behaviour of parasites during invasion (Appendix Fig S3, see also [24]. Combining live cell and super-resolution microscopy, we were able to identify two situations in WT parasites: in the majority of cases (76%), we detected F-actin forming a ring coinciding with the TJ, while in ~24% of all cases, no significant accumulation was detectable, as confirmed by intensity plots (Fig 3A–C, Appendix Fig S4, Movie EV6). Importantly, genetic interference with actin dynamics led to constrictions/TJ devoid of F-actin at the junction, as seen in the case of *adf* cKD (Fig 3A and B, Movie EV6), demonstrating that F-actin accumulation at the junction requires correctly regulated F-actin dynamics, as suggested previously [31]. Previously, it was shown that WT parasites can enter the host cell through static or capped TJ depending on whether the TJ is respectively firmly or loosely anchored [4]. Analysis of parasites entering the host cell via capping shows F-actin at the TJ (Fig 3A and B, Appendix Fig S4, Movie EV6), confirming the finding that both invasion scenarios rely on the parasite F-actin-based traction force [4].

Although we were unable to accurately quantify F-actin accumulation at the junction, we consistently observed that it increased during the invasion event (Fig 3A, Movie EV6). At the onset of invasion, only a relatively faint F-actin signal could be detected at the apical tip of the parasite, its intensity increasing during invasion, culminating when the parasite invaded to ~2/3. Collectively, while these observations fit well with the contribution of F-actin to the traction force applied at the TJ, the amount of invasion for which we could not detect F-actin at the TJ (25% show no F-actin ring) is close to the fraction of residual invasiveness detected for mutants for the acto-myosin system, such as shown in independent studies for mutants for *act1, mlc1, myoA, myoH, GAP45 and adf* [11,36–38]. Together, these observations suggest that interference with either MyoA or actin dynamics results in a reduction of parasite invasion in line with the absence of an F-actin ring at the entry point.

While time-lapse analysis of invasion events allowed documenting the dynamics of F-actin at the junction and posterior pole of the parasite, this analysis remained limited by insufficient resolution. Therefore, we decided to obtain higher resolution images using fixed assays to confirm the observed variability in F-actin accumulation at the junction. We quantified F-actin at the junction and the posterior pole using a modified red-green assay based on differential immunolabelling [39]. Invading parasites expressing Cb-EmeraldFP were labelled with SAG1 (non-invaded part of the parasite) and Ron2 (TJ-marker). We confirmed that in all cases, parasites entered the host cell via a TJ, as indicated by positive Ron2 staining (Fig 3D).

In good correlation with live imaging analysis, we detected a F-actin ring at the TJ in ~80% of RH parasites, leaving ~20% of the events where the presence of an F-actin ring remained questionable in fixed samples (Fig 3D and E). In contrast, in all invading parasites, a posterior accumulation of F-actin was apparent (orange arrow, Fig 3D and F). In the case of *adf* cKD, which showed a reduced invasion rate of ~20% compared to WT parasites, no F-actin ring at the junction was obvious in any invasion event (red arrowhead, *adf* cKD panel, Fig 3D and E), while the posterior accumulation was still maintained (orange arrow, Fig 3D and F). As expected, no F-actin at the junction or at the posterior pole of the parasite was detectable in *act1* cKO parasites and only a cytosolic signal for Cb-EmeraldFP was apparent (Fig 3A, D, E and F). Similar to *adfc*KD, an overall invasion rate of ~20% compared to WT parasites was detected for *act1*cKO parasites, as described previously [9], despite the posterior, aberrant morphology described previously [7,9] (green arrow, Fig 3D). Interestingly, in the case of *myoA*KO parasites, we found (similar to *adf*KD *and act1*cKO*)* an invasion rate of ~20% compared to WT parasites, confirming previous results [9,16,40]. However, disruption of *myoA* resulted in loss of F-actin accumulation at both the junction site and the posterior pole, demonstrating that MyoA is the main motor required for retrograde F-actin flow during host cell invasion. Interestingly detectable F-actin appears to be equally distributed within the cytosol of the parasite and at the periphery of the parasite (Fig 3D–F), confirming the data obtained for extracellular parasites (Fig 1).

Together, these data confirm that an intact acto-myosin system is required for efficient host cell invasion and that the accumulation of F-actin at the posterior pole can occur independently of its accumulation at the junction, suggesting two independent processes, both dependent on MyoA-motor activity and F-actin dynamics. At this point, our analysis does not allow to pinpoint how F-actin dynamics driven by a Glideosome localised between the IMC and PM is coordinated and connected with the changes in F-actin dynamics observed within the cytosol of the parasite. One potential explanation could be that both processes are regulated by the same signalling cascades, which is supported by the actin flow analysis shown above (Fig 1).

In analogy to other eukaryotes migrating through a constricted space, where the nucleus represents a major obstacle that needs to be deformed, squeezed and pushed through the constriction [35], we speculated that the nucleus of the parasite represents a major limitation for invasion that needs to be forced through the junction.

## Posterior localised F-actin assists nuclear entry

We next simultaneously visualised F-actin dynamics and the position and shape of the nucleus using time-lapse imaging (Fig 4). We confirmed that the accumulation of F-actin at the junction became most prominent, once the nucleus reached the junction (orange arrow, Fig 4A, Movie EV7). Interestingly, in the vicinity of F-actin located at the parasite posterior pole, we observed a cup-like meshwork around the nucleus close to the junctional F-actin ring (blue arrow, Fig 4A, Movie EV7). Once the nucleus reached the junction, it became constricted and deformed when the F-actin ring is visible, (see red arrowhead 0:15–0.21, Fig 4A and B). When no F-actin ring was detectable (Fig 4B), the shape of the nucleus remained relatively constant (red arrows, Fig 4D and E). However, independently of F-actin ring formation, the posterior pole of the parasite contracted (blue arrows in Fig 4A and D; Movie EV7) prior or coinciding with nuclear entry, which was also confirmed by measuring

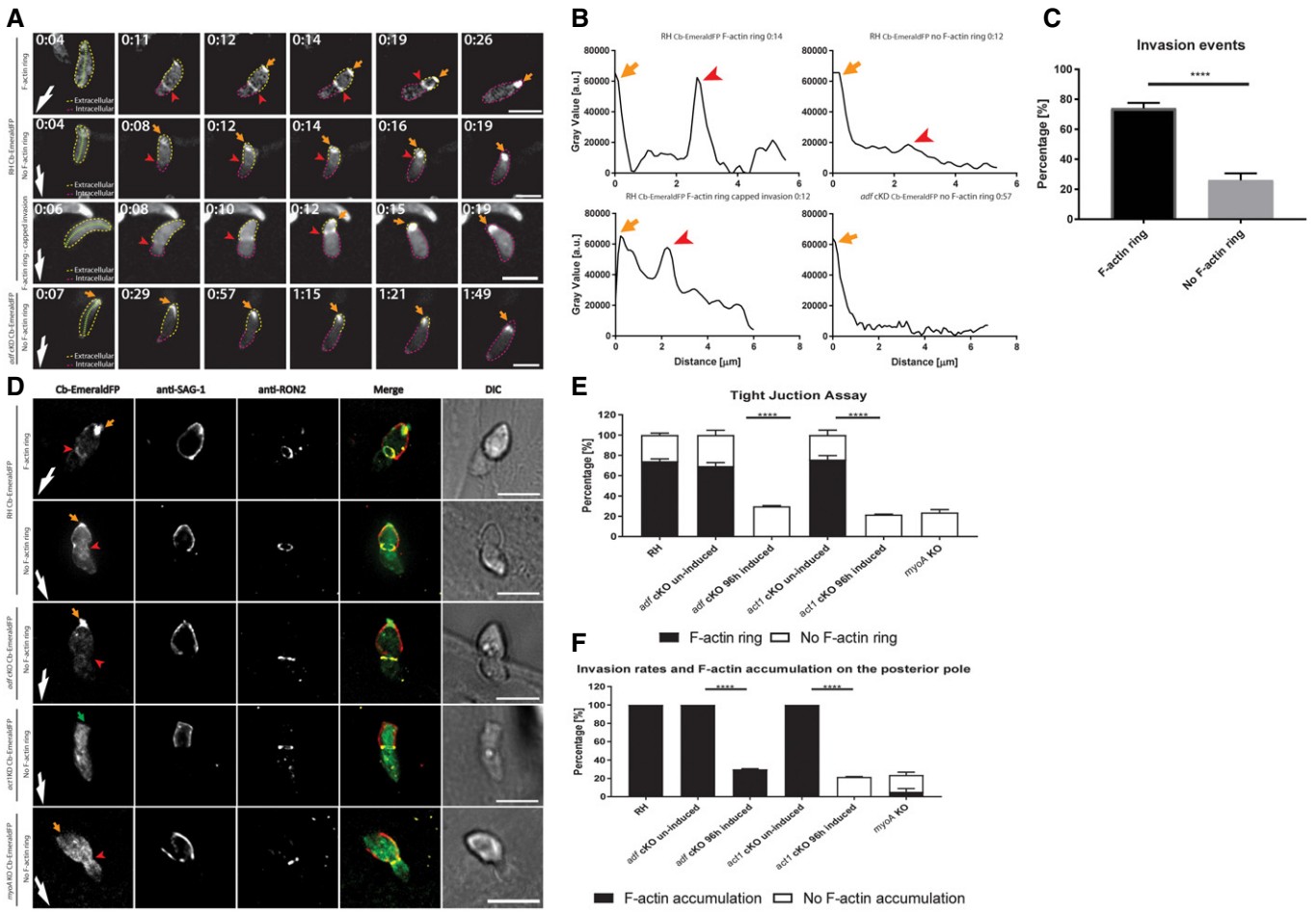

**Figure 3.** F-actin dynamics during *Toxoplasma gondii* invasion during fixed and live imaging.

A   Time-lapse analysis depicting invading RH *adf*cKD parasites expressing Cb-EmeraldFP. A F-actin ring can be observed at the TJ in the majority of cases (red arrowheads, first panel). Invasion can also occur without formation of an F-actin ring (second panel). During capped invasion, an F-actin ring is formed (third panel). Invasion of *adf* cKD parasites occurs without formation of an F-actin ring (fourth panel). F-actin accumulation is observed at the posterior pole of the parasite in all cases (orange arrows). Green segmented lines depict the area that was measured for generation of intensity plot profiles shown in (B). Yellow and purple dotted lines indicate the extra- and intracellular part of the parasite, respectively. The time lapse is presented in Movie EV6.

B   Intensity plot profiles of the movies shown in (A). Depicted plots correlate to half-invaded parasites. In the case of F-actin ring formation, two distinct peaks are detected, correlating to the F-actin ring at the TJ (red arrowhead) and the posterior pole of the parasite (orange arrow). In the case of no F-actin ring, no accumulation at the TJ can be observed; only the peak correlating to posterior F-actin can be detected. For *adf* cKD, only one peak can be seen corresponding to the posterior pole of the parasite.

C   Quantification of F-actin accumulation at the TJ for RH Cb-EmeraldFP parasites as detected in time-lapse analysis. The values are expressed as percentage. 28 total parasite invasion events were captured across three different biological replicates. Two-tailed unpaired *t*-test analysis was performed. ****$P < 0.0001$.

D   Representative immunofluorescence assays on indicated parasite strains during host cell penetration. Parasites were labelled with indicated antibodies. SAG1 staining was performed prior to permeabilisation to stain only the extracellular portion of the parasite. Antibodies against Ron2 were used to stain the TJ. Red arrowhead: position of the TJ. Orange arrow: posterior pole of the parasite.

E   Tight junction assay on fixed samples (as shown in (D)) depicting invasion rates of indicated parasites. While in the majority of control parasites, F-actin can be detected at the junction (~80%), no junctional ring can be detected in ~20% of all invasion events. In contrast, *adf* cKD, *act1* cKO or *myoA* KO only shows an overall invasion rate of ~20% compared to controls. In these cases, junctional F-actin accumulation could never be detected. Note that the number of total invasion events for *adf* cKD, *act1*cKO and *myoA*KO correlates to the number of invasion events, seen for RH without F-actin ring formation.

F   Quantification of posterior F-actin accumulation during invasion of indicated parasite lines.

Data information: Error bars represent standard deviation. White arrows point to direction of invasion. Scale bars represent 5 µm. One-way ANOVA was performed for each graph in (E, F). ****$P < 0.0001$.

Source data are available online for this figure.

the diameter of the posterior pole of the parasite during the invasion process (Fig 4C and F).

We then imaged the nucleus of *myoA*KO parasites over their residual host cell invasion (Fig 4G). Since the MyoA-motor-complex is known to generate the required traction force at the TJ [10], we interpret these data as if optimal entry by WT parasites results from a coordinated mechanism, requiring the Glideosome, acting below the surface of the parasite [6], together with

cytosolic localised F-actin accumulating at the posterior pole and surrounding the nucleus. In good support, when no F-actin was detectable at the junction, indicating independence of traction force generated at the junction, posterior accumulation of F-actin adjacent to the nucleus was still seen during invasion (Fig 4D–F, Movie EV7).

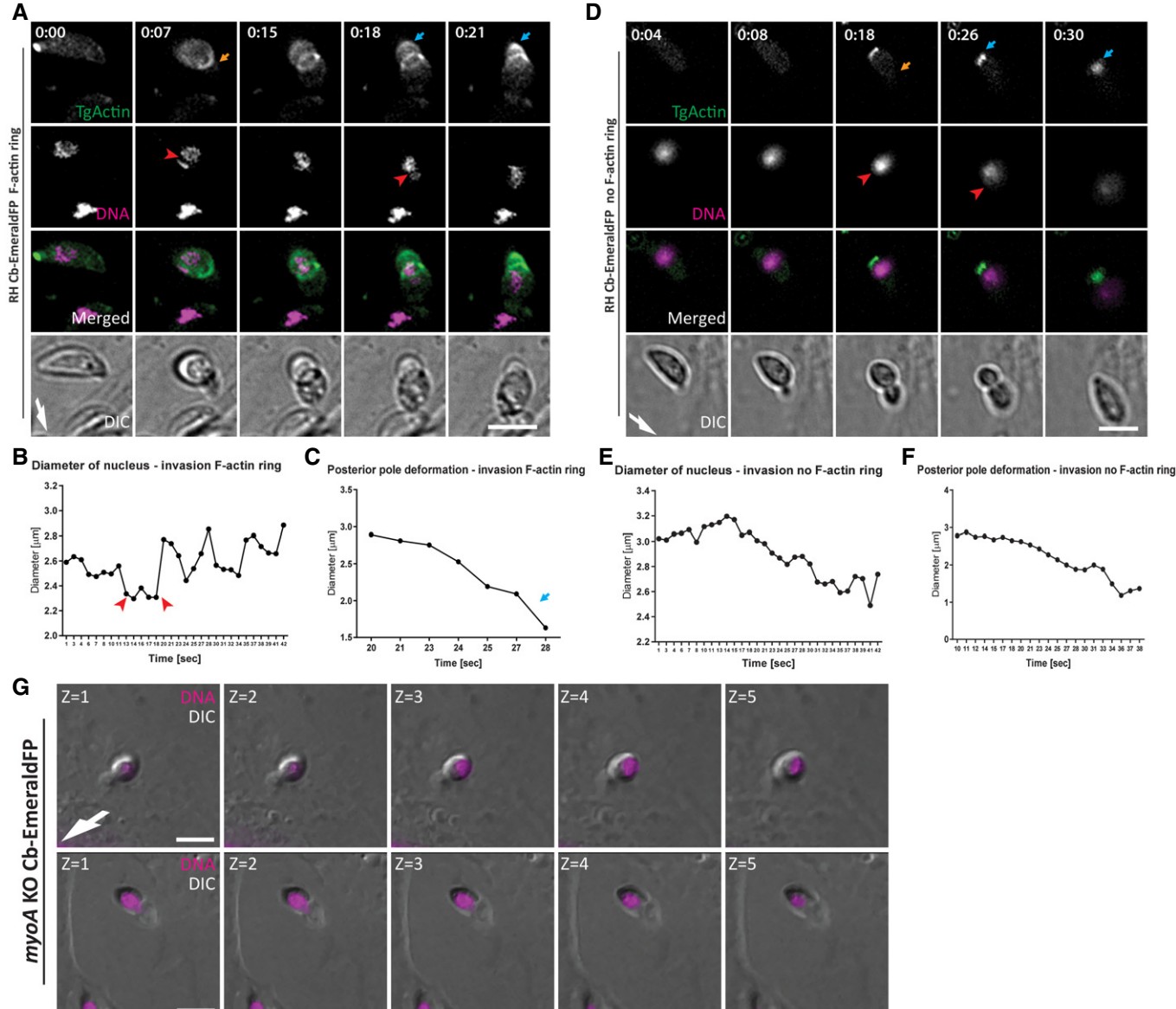

**Figure 4. Time-lapse analysis of F-actin and nuclear dynamics during host cell invasion.**

A   Time-lapse analysis of invading RH Cb-EmeraldFP parasites. During penetration, an F-actin ring is formed at the TJ (orange arrow). The nucleus (purple) is squeezed through the TJ (red arrowhead), and posterior F-actin appears to be directly connected to the nucleus and TJ (blue arrow). The time lapse is presented in Movie EV7.

B   Analysis of nucleus deformation during live imaging shown in (A). The nucleus constricts while it is passing through the TJ (red arrowheads).

C   The posterior end of the parasite deforms during the invasion process. F-actin is accumulated during invasion, with the posterior end contracting (blue arrow).

D   Time lapse stills depicting RH Cb-EmeraldFP parasites invading in the absence of detectable F-actin at the junction (orange arrow). The nucleus (purple) is squeezed through the TJ (red arrowhead) during invasion. Actin accumulation at the posterior pole of the parasite is strongly detected in all cases (blue arrow). The time lapse is presented in Movie EV7.

E   Analysis of nucleus deformation during live imaging shown in (D). The nucleus appears to not show constriction once it is passing through the TJ when the F-actin ring is not present.

F   The posterior end of the parasite also deforms during the invasion process.

G   Z-stack gallery depicting mid-invading *myoA* KO parasites. Note that the nucleus is located at the back during invasion events.

Data information: Numbers indicate minutes:seconds. Scale bar represents 5 μm. White arrow points to direction of invasion.
Source data are available online for this figure.

## F-actin forms a continuous network between the junction and the posterior accumulation that surrounds the nucleus

We applied super-resolution structured illumination microscopy (SR-SIM) to analyse the distribution of F-actin in WT parasites that were at the mid of invasion and found consistently that the nucleus of the parasite is surrounded by F-actin that formed a continuous meshwork connected to the posterior pole of the parasite (Fig 5A–D). Importantly, this meshwork could be observed independently of F-actin accumulation at the junction but broke down upon disruption of *adf* and was significantly reduced in the case of *myoA*KO parasites (Fig 5A and D Movie EV8). The perinuclear F-actin meshwork formed a continuum with the posterior accumulation (Fig 5B). Finally, 3D models demonstrated that F-actin is perinuclear and can be observed in ~60% of all cases in WT parasites, while it is significantly reduced in *myoA*KO (~20%) and absent in *adf*KD parasites (Fig 5D).

In most migrating cells, the nucleus is positioned in the back, which requires the coordinated action of posterior F-actin as well as anterior microtubules, resulting in positioning of the nucleus by a pull-and-push mechanism driven by the interplay of an acto-myosin complex, actin-nucleus interface proteins and other cytoskeletal structures [28]. In the case of *T. gondii,* the subpellicular microtubules are polymerised from microtubule organising centres at the apical tip, aligned on the cytosolic side of the IMC, where they directly interact with gliding-associated membrane proteins, which are components of the Glideosome (GAPMs [22]). This subpellicular microtubules form a basket that covers ~2/3 of the parasite length. Furthermore, it was previously suggested that parasite F-actin can interact with the subpellicular microtubules [19,21], leading to the question if the basally localised meshwork consisting of F-actin might interact with the apically localised subpellicular microtubules during invasion. Therefore, we assessed the location of microtubules (MTs) during invasion by the use of SiR-tubulin [41]. The MTs cover 2/3 of the parasite body creating a rigid structure starting from the apical tip (Fig 5E–M; Movie EV9). During invasion, this MT structure appears to be significantly deformed, with several examples detected, where MTs seem to colocalise in a positive correlation with junctional F-actin (white arrows in Fig 5; Pearson's coefficient and white marker in 3D rendering Appendix Fig S5). Furthermore, MTs appear frequently bent and distorted at the junction. Interestingly, this polarisation with F-actin at the posterior end and microtubules at the leading edge is maintained throughout the invasion process, from early to late stage (Fig 5). In some cases, the MTs extended far beyond the 2/3 parasite body, aligning the nucleus (white arrow, Fig 5). The microtubules closely align to the part of the nucleus that enters through the TJ (orange arrowheads), while F-actin closely aligns to the posterior part of the nucleus (green arrowheads), resulting in a highly polarised cytoskeletal organisation around the nucleus (Appendix Fig S5; Movie EV9). It is plausible that akin to other eukaryotes, a coordinated action of microtubule and F-actin-based mechanisms supports nuclear entry and deformation through the TJ (Appendix Fig S6A and B). In good agreement with this hypothesis, a positive correlation between microtubules and actin colocalisation during the invasion process is observed for WT parasites, but not *myoA*KO or *adfc*KO (Figs 5 and Appendix Fig S6A). This basic organisation is also maintained in those cases, where no F-actin accumulation is detectable at the junction (Fig 5G).

Upon interference with actin dynamics (*adfc*KO) or the Glideosome (*myoA*KO), this organisation is disrupted (Appendix Fig S6B). In case of ADF depletion, posterior accumulation of F-actin is still apparent (green arrowheads, Fig 5K–M), but no F-actin meshwork around the nucleus, connecting to subpellicular microtubules, is detectable (orange arrowheads, Fig 5K–M), while microtubules and actin colocalise at the apical tip of the parasite (Appendix Fig S6B). In contrast, disruption of MyoA results in a dominant arrangement of F-actin throughout the cytosol with no accumulation at the posterior pole of the parasite or formation of perinuclear F-actin (Fig 5H–J), again suggesting that there is a tight coordination of Glideosome function in between the PM and IMC and the formation of the cytosolic localised meshwork around the nucleus. It is also noteworthy that F-actin accumulates in the region of the TJ (green arrowheads) when compared to the rest of the cell, where it shows a good colocalisation with microtubules (Appendix Fig S6B).

In summary, the combined data from fixed invasion assays and SR-SIM imaging allow documenting that the parasite's nucleus is surrounded by a F-actin meshwork and that its formation and coordination might depend on subpellicular microtubules. Here, F-actin appears to adapt a cytoskeletal conformation designed to protect, deform and push the nucleus inside the host cell and we propose that during invasion, the Glideosome provides the traction force for general movement of the parasite through the TJ, while perinuclear actin is important in order to facilitate nuclear entry.

## Perinuclear F-actin is detected during invasion of Plasmodium merozoites

While *T. gondii* is a relevant model to address general questions in apicomplexan biology, we previously observed differences in the role of F-actin during the asexual life cycles of *T. gondii* and *P. falciparum*, the causative agent of human malaria. For instance, in case of *P. falciparum*, invasion is dependent on parasite F-actin [42], while in the case of *T. gondii,* residual invasion can be observed. To analyse the role of F-actin during merozoite invasion, we expressed CB-Halo and Cb-EmeraldFP in *P. falciparum* and validated their efficiency to analyse F-actin localisation and dynamics in this parasite (see also [25]). Invasion assays and localisation analysis of F-actin during merozoite invasion showed a similar distribution of F-actin at the junction, around the nucleus and at the posterior pole of the parasite in 100% of all analysed images ($n > 100$) (Fig 6A and C). We also analysed F-actin localisation during the invasion of *P. knowlesi* merozoites using a previously described antibody [15] that preferentially recognises *P. falciparum* F-actin and obtained identical results (Fig 6B). Fixed immunofluorescence imaging of merozoites in the process of invasion was analysed for the distribution of F-actin. As with *T. gondii,* F-actin can be detected at the junction, at the onset of invasion (Fig 6) pointing to a similar strategy between apicomplexan parasites. Importantly, super-resolution imaging of F-actin in merozoites in different stages of invasion clearly highlights the accumulation of F-actin at the posterior pole and in direct contact with the nucleus (Fig 6C). Quantification of invasion events demonstrated that in over 80% of cases, a close actin–nucleus association can be detected (Fig 6D) and, in over 90% of cases, actin was accumulated at the TJ (Fig 6D).

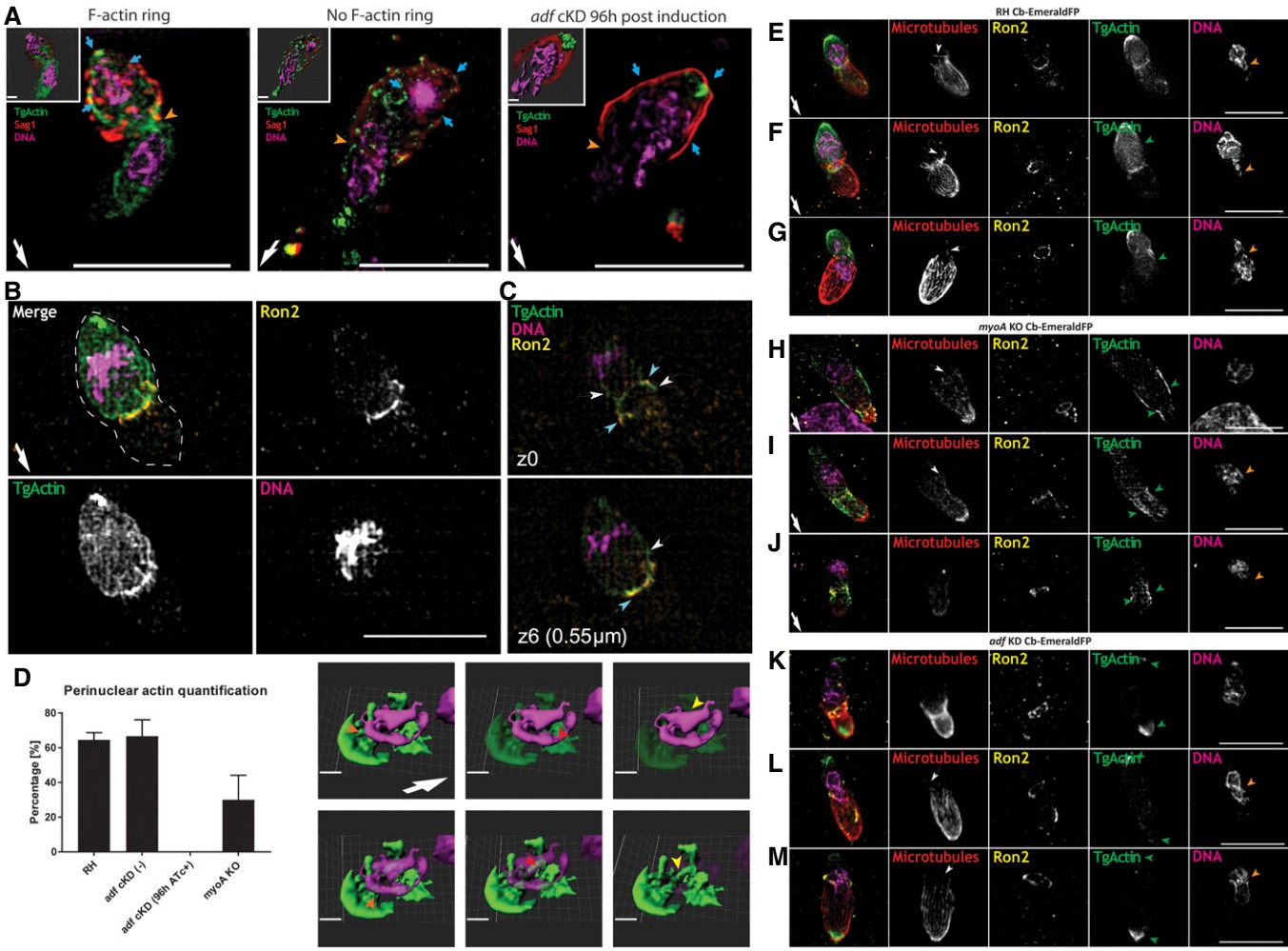

**Figure 5. Super-resolution microscopy demonstrates formation of a F-actin cage around the nucleus during invasion.**

A    SR-SIM images depicting invading parasites with and without the presence of the F-actin ring at the tight junction and invading *adf* cKD Cb-EmeraldFP parasites. SAG1 staining (in red) was performed prior to permeabilisation to specifically label extracellular part of the parasite. Note the accumulation of actin (blue arrow) around the nucleus and the posterior pole of the parasite, irrespective of F-actin ring formation at the TJ (orange arrowhead). 3D rendering is shown for each presented case (inlet). 3D rendering is presented in Movie EV8.

B    SR image of an invading parasite before nuclear entry. F-actin forms a continuous meshwork around the nucleus that appears connected to the posterior pole.

C    F-actin localisation in relation to the TJ during invasion. F-actin (white arrow) is in close vicinity to the Ron2 ring (light blue arrow) during invasion.

D    Quantification of perinuclear actin in indicated parasites. Parasites were counted by assessing Cb-EmeraldFP signal as an actin marker around or in the vicinity of the nucleus. Still images to the right represent what was considered as F-actin–nucleus association. The orange arrowhead points to F-actin surrounding the nucleus in the posterior end of the parasite, and the red and yellow arrowhead shows F-actin in the vicinity of the nucleus, hinting at a close interaction between the two. Three biological replicates were analysed with a minimum of 25 parasites per replicate.

E–G    SR-SIM images showing three stages of invading wt parasites. SiR-tubulin staining for microtubules (in red) was performed prior to fixation to specifically label microtubules in the parasite. During invasion, the microtubules (MT) are deformed at the TJ area (white arrowhead). The accumulation of actin (green arrow) at the posterior forms a F-actin meshwork. The nucleus is deformed when it enters the TJ area (orange arrowhead). These images are expanded with additional panels in Appendix Fig S5. 3D rendering is presented in Movie EV9.

H–J    In the case of the *myoA* KO strain, the MTs also deform when passing through the TJ (white arrowhead). However, F-actin is not concentrated at the posterior pole or the TJ, but evenly distributed within the cytosol of the parasite, with some peripheral location.

K–M    In case of *adf*KD, MTs are deformed at the TJ area (white arrowhead). F-actin is accumulated at both ends of the parasite (green arrowheads), with no accumulation at the TJ.

Data information: Error bars represent standard deviation. Scale bar represents 5 μm for SR-SIM images and 1 μm for 3D models. White arrow points to direction of invasion.

Source data are available online for this figure.

# Discussion

Host cell invasion by apicomplexan parasites occurs through a tight junctional ring (TJ), established by the parasite upon secretion of its rhoptry content [3]. Previous studies demonstrated that the TJ is anchored to the host cell cortex via its interaction with host cell factors [5,43] thereby acting as an anchor for traction force exerted by the parasites acto-myosin system [10]. In addition, compressive

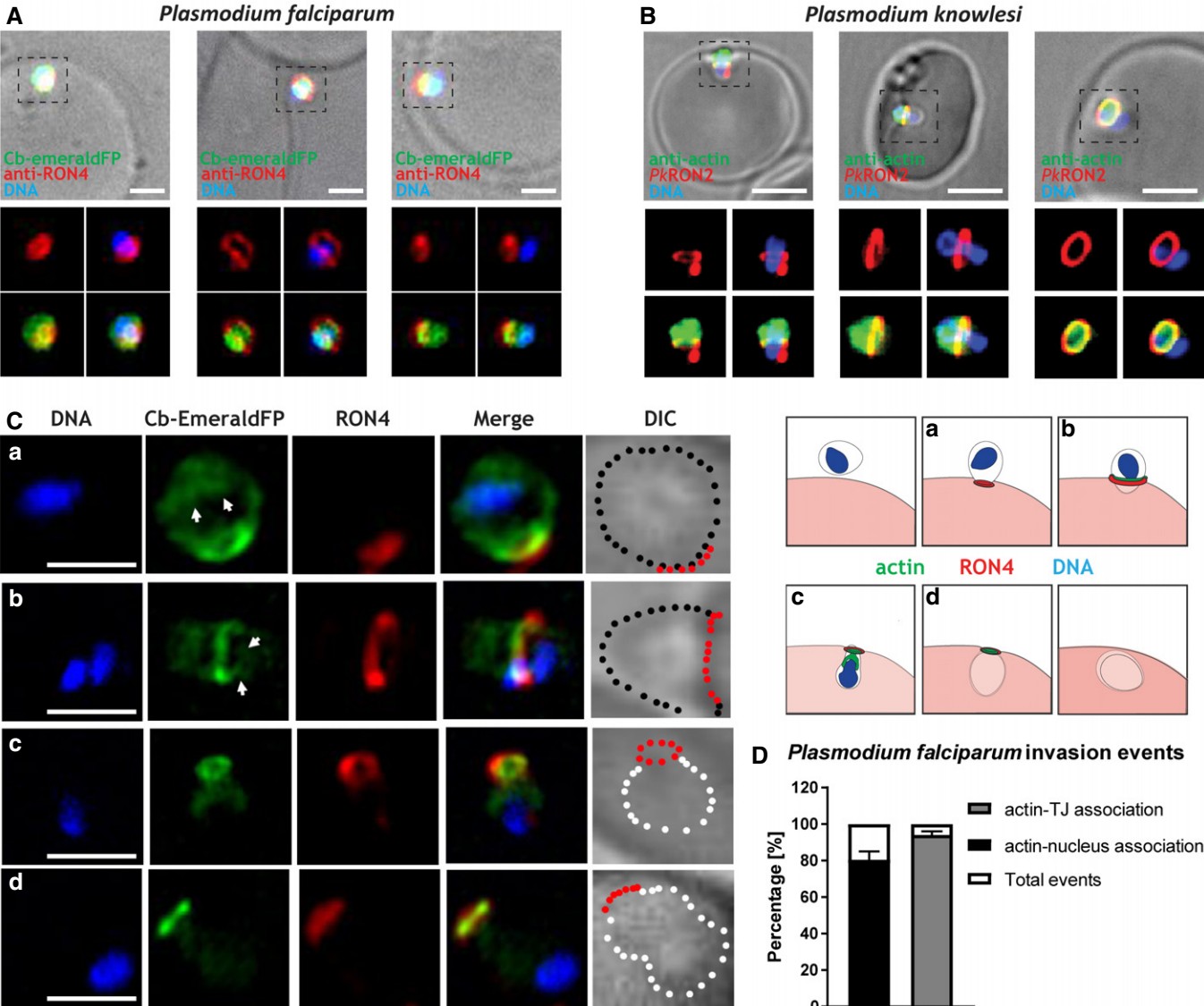

**Figure 6. *Plasmodium* F-actin localisation during invasion.**

A   Different invasion stages of *P. falciparum* merozoites expressing Cb-EmeraldFP. DAPI labels the nucleus (DNA), Cb-EmeraldFP labels actin filaments (CB-EME, green), and anti-RON4 labels the junction (red). Left panel: onset of invasion. Middle panel: a merozoite in the middle of invasion; note the constriction of the nucleus (blue) as it passes through the RON4 junction and F-actin colocalising with it (green). Right panel: a merozoite completing invasion; note the F-actin ring beyond the RON4-junction and actin filaments surrounding the nucleus.

B   IFA showing various stages of invasion of erythrocytes by *P. knowlesi* merozoites. DAPI labels the nucleus (DNA), anti-actin antibody labels actin filaments (anti-actin, green), and the invasion junction is marked with a C-terminally tagged *Pk*RON2 mCherry-HA (red). Left panel: a merozoite beginning invasion; note the formation of an F-actin ring (green) behind the *Pk*RON2 junction (red). Middle panel: a merozoite in the middle of the invasion event; note the constriction of the nucleus (blue) as it passes through the *Pk*RON2 junction and F-actin colocalising with it (green). Right panel: a merozoite completing the process of invasion; note the F-actin ring beyond the *Pk*RON2-junction and actin filaments colocalising with the nucleus.

C   Super-resolution images showing various stages of invasion of erythrocytes by *P. falciparum* merozoites expressing Cb-EmeraldFP. Left panel: DAPI labels the nucleus (blue), Cb-EmeraldFP labels actin filaments (CB-EME, green), and the invasion junction is marked with an anti-RON4 antibody (red). Brightfield images have been marked with red dots depicting the junction, black dots depicting merozoite boundary still outside the erythrocyte and white dots depicting a merozoite that has penetrated the host cell. Top panel shows an attached merozoite beginning the process of invasion, the second panel during the process of invasion, and the bottom two panels show merozoites completing the process of invasion. Right panel: schematic depicting merozoite invasion into red blood cells. Letters depict the corresponding event in the model (left) and microscopy pictures (right). White arrows depict actin filaments that colocalise with the nucleus.

D   Quantification of *Plasmodium falciparum* invasion events categorising actin-TJ and actin-nucleus association events. 121 events were counted in total.

Data information: Error bars represent standard deviation. Scale bar represents 1 μm for *Plasmodium falciparum* and 3 μm for *Plasmodium knowlesi*.
Source data are available online for this figure.

forces, caused by the host cell at the TJ, require the deformation of the parasite during invasion, which appears to be counterbalanced to ensure integrity of the parasite [10]. In fact, the invasion of host cells through the TJ appears in many aspects very similar to the migration of eukaryotic cells through a constricted environment, with impressive morphological changes observed at the point of constriction [44]. Here, the nucleus represents a physical barrier for migration that requires to be modulated, deformed and protected in order to be squeezed through the constriction [35]. During migration through a constricted space, rapid actin nucleation around the nucleus facilitates nuclear deformation and protection. Furthermore, proteins that interface actin and the nucleus such as the family of LINC (Linker of Nucleoskeleton and Cytoskeleton) proteins allow the formation of intricate perinuclear F-actin structures connected to microtubules and other cytoskeletal elements that support nucleus translocation, mechanotransduction and protection [45,46].

In apicomplexan parasites, a similar function of F-actin was initially ruled out based on the assumption that F-actin acts exclusively within the narrow space between the PM and IMC, where the Glideosome is localised [2]. However, the finding that the majority of F-actin dynamics appears to occur within the cytosol of the parasite in extracellular parasites, and the fact that host cell invasion is significantly slowed down once the parasite's nucleus reaches the TJ and is usually blocked at this stage in the case of mutants for the acto-myosin system, led us to the hypothesis that parasite F-actin serves a second role during invasion, namely the facilitation of nuclear entry, akin to other eukaryotes, where a squish and squeeze model has been proposed [28].

To test this hypothesis, we performed a careful analysis of WT parasites and mutants for core components of the acto-myosin system and correlated their phenotypes with the observed changes in F-actin dynamics in extracellular and invading parasites, using quantitative imaging approaches allowing us to describe the dynamics of F-actin during host cell invasion. We used a combination of real-time and super-resolution microscopy to demonstrate that the nucleus is indeed a major obstacle that needs to be overcome for successful invasion. In good agreement, mutants of the acto-myosin system demonstrate an extended pause during invasion, as soon as the nucleus reaches the junction. In all cases, parasites accumulate F-actin at the posterior pole during invasion, which forms a continuous meshwork with perinuclear F-actin. In contrast, F-actin accumulation at the junction is not always observed during the invasion process, suggesting that the Glideosome activity is fine-tuned, depending on the force required for host cell penetration. Importantly, F-actin accumulation at the posterior pole can occur independently from F-actin accumulation at the junction, suggesting the action of two independent pools of F-actin that contribute to invasion and nuclear entry. Finally, we demonstrate that parasite F-actin and subpellicular microtubules form a highly polarised cell during invasion, with the subpellicular microtubules at the leading edge and F-actin at the posterior pole, raising the question if a tight and controlled coordination of these two cytoskeletal elements facilitates host cell invasion.

While the interplay between the F-actin and microtubule cytoskeleton of the parasite needs to be further analysed, it is interesting to note that deletion of *myo*A, the core motor of the Glideosome [40], results in complete loss of F-actin polarity and an even distribution of F-actin in the cytosol of the parasite, with some peripheral accumulation during motility and invasion. This indicates

a tight interlinkage of F-actin dynamics occurring in between the PM and IMC and controlled by the action of the Glideosome with the F-actin dynamics observed within the cytosol of the parasite. This is also reflected, by the fact that F-actin dynamics originating from the apical tip of the parasite, potentially by Formin-1 and translocated via MyoH [11] and close to the Golgi by Formin-2 [11,25] forms a continuous network that appears to be interlinked and demonstrates bidirectional F-actin flow. Interestingly, stimulation of calcium signalling with either A23187 or BIPPO resulted in preferential F-actin flow towards the basal end of the parasite. Surprisingly, F-actin dynamics in unstimulated *myoA*KO parasites is identical to WT parasites, but upon stimulation, F-actin relocation to the basal pole appears to be blocked and remains "stuck" in the first half of the parasite, demonstrating that the Glideosome is required for efficient relocation of the whole F-actin pool and that both processes are tightly interlinked. Although a more in depth analysis of the myosin machinery could provide more evidence of the interplay of actin dynamics, only MyoA and MyoH have been implicated in invasion. However, MyoH mutants were shown to be unable to initiate invasion [23], making it impossible to study the role of this myosin for parasite and nuclear entry.

Analysis of host cell invasion demonstrated that F-actin can be found at the TJ in most cases and—as expected—disruption of MyoA, ADF or Act1 results in highly reduced invasion rates, where no F-actin can be detected at the TJ. It remains to be seen, if the F-actin ring is formed in order to stabilise the junction to counteract pressure exerted by the host cell [10] or if the activity of the Glideosome can be modulated, depending on the necessity to provide the traction force to facilitate entry of the parasite. Collectively, while these observations fit well with the contribution of F-actin to traction force applied at the TJ, the amount of invasion for which we could not detect F-actin at the TJ (25% show no F-actin ring) equalled the fraction of residual invasiveness detected for mutants for the acto-myosin system, such as shown in independent studies for mutants for *act1, mlc1, myoA, myoH, GAP45* and *adf* [7,9,11,31,36–38,40].

A similar situation is seen for the formation of filamentous actin structures in close proximity to the nucleus during host cell invasion, with most parasites demonstrating the formation of an F-actin meshwork around the nucleus appearing as perinuclear actin structures. Again, disruption of the acto-myosin system causes disruption of this meshwork, demonstrating a tight interplay between Glideosome function and formation of perinuclear actin. In contrast, in all cases, F-actin is seen to accumulate at the basal pole of the parasite during invasion, which is connected to the perinuclear meshwork (if formed during invasion). Finally, live imaging analysis indicates that the basal end of the parasite contracts during host cell entry, potentially forcing the nucleus through the junction.

In most migrating cells, the nucleus is positioned in the back, which requires the coordinated action of posterior F-actin as well as anterior microtubules, resulting in positioning of the nucleus by a pull-and-push mechanism driven by the interplay of an acto-myosin complex, actin-nucleus interface proteins and other cytoskeletal structures such as intermediate filaments and microtubules [28]. In the case of *T. gondii*, the subpellicular microtubules are polymerised from microtubule organising centres at the apical tip, aligned on the cytosolic side of the IMC, where they directly interact with components of the Glideosome, such as the gliding-associated membrane

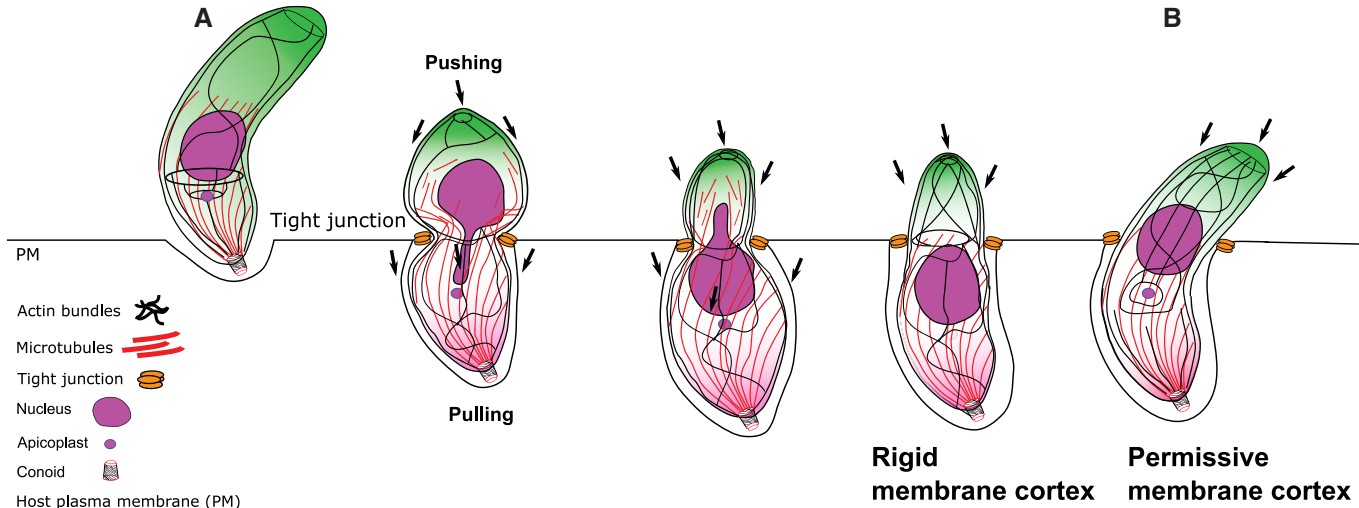

**Figure 7.  Model of the proposed nuclear squeeze mechanism during apicomplexan invasion.**

A   Once the parasite attaches to the surface of a host cell, F-actin strongly accumulates at the posterior pole and at the apical end. During penetration, the junctional complex is formed that contributes to the attachment and stabilisation of the parasite to the host cell in the TJ. F-actin at the TJ provides traction force and stability for nuclear entry, while posterior F-actin provides contraction force to allow nuclear entry. At the same time, microtubules might facilitate nuclear entry by pulling. We propose that the nucleus is squeezed through the TJ by a pushing–pulling mechanism controlled by actin and potentially microtubules.

B   In some cases (e.g. due to more permissive host cells or upon modulation of F-actin dynamics), a F-actin ring at the junction is not required/formed and the nucleus can enter the host cell by the action of posterior accumulated F-actin.

proteins (GAPMs [22]). Indeed, it was previously suggested that parasite F-actin can interact with the subpellicular microtubules [19,21], leading to the question if the posterior localised meshwork consisting of F-actin might interact with the apically localised subpellicular microtubules during invasion to form a highly polarised cellular network, as seen in Figs 5 and Appendix Fig S5. Further studies are required to analyse the linkage of these two cytoskeletal elements as subpellicular microtubules could function as stabilisers of cell shape but could be also involved in force generation during invasion.

Together, these results permit us, for the first time, to reconcile discrepancies in the phenotypic invasion characteristics of apicomplexan parasites with F-actin imaging data and to propose a new model for actin's contribution during host cell invasion, in particular nuclear entry through the TJ when F-actin accumulation is present in the TJ (Fig 7A). In other cases where F-actin was not detected, we hypothesise a more permissive host cell membrane cortex is encountered upon invasion that facilitates parasite's entry, thus reducing the force required for entry (Fig 7B). While the IMC-localised acto-myosin motor complex provides traction force at the TJ to initiate and facilitate rapid cell invasion, the entry of the parasite nucleus into the host cell appears to be a major limiting step that requires direct action of a cytosolic acto-myosin system, possibly also involving the subpellicular microtubules in order to be deformed, protected and pushed into the host cell (Fig 7A).

## Materials and Methods

### Plasmid construction

The same procedure was carried out as described previously [24]. The Cb-EmeraldFP plasmid consists of a sequence encoding actin

chromobody (Cb) from Chromotek followed downstream by an in-frame sequence encoding EmeraldFP. The vector backbone contains a *DHFR* promoter for protein expression without drug resistance genes. To create Cb-EmeraldFP, the EmeraldFP coding sequence was amplified using primers (EmeraldFP-F: atgcaccggtatgggactcgt-gagcaaggg and EmeraldFP-R: atgccttaagttacttgtacagctcgtcca). The EmeraldFP PCR product plasmid was subcloned using traditional restriction digestion/ligation protocols.

### Culturing of parasites and host cells

Human foreskin fibroblasts (HFFs) (RRID: CVCL_3285, ATCC) were grown on tissue culture-treated plastics and maintained in Dulbecco's modified Eagle's medium (DMEM) supplemented with 10% foetal bovine serum, 2 mM L-glutamine and 25 mg/ml gentamicin. Parasites were cultured on HFFs and maintained at 37°C and 5% $CO_2$. Cultured cells and parasites were regularly screened against mycoplasma contamination using the LookOut Mycoplasma detection kit (Sigma) and cured with Mycoplasma Removal Agent (Bio-Rad) if necessary.

### *Toxoplasma gondii* transfection and selection

To generate stable Cb-EmeraldFP expressing parasites, $1 \times 10^7$ of freshly released RH Δ*hxgprt*parasites and *adf* cKO were transfected with 20 μg DNA by AMAXA electroporation. Transfected parasites were then sorted using flow cytometry with a S3 Cell Sorter (Bio-Rad, Hercules, CA, USA).

To create Myo-SNAP vector, PMyoA-Ty-MyoA [9] was used as a template backbone and the Ty sequence replaced in frame with a sequence encoding SNAP from plasmid template SNAP (NEB) amplified with primers FWSNAP 5'TAATGAATTCCGACAAAA

TGGACAAAGACTGCGAAATGA3′ and RVSNAP 5′TAATATGCATAC
CCAGCCCAGGCTTGCCCAG3′.

## Merozoite isolation

PkRON2*mCherry-HA A1-H.1 merozoites were isolated as
described [47]. In brief, ring-stage parasites were tightly synchro-
nised to a 3.5 h and allowed to develop to schizont stage. Schi-
zonts were separated from uninfected erythrocytes by MACS
magnet separation (CS column, Miltenyi Biotec). Parasites were
returned to complete media at 37°C and incubated with 10 μM E64
(Sigma-Aldrich) for < 4 h. Schizonts were pelleted (850 $g$, 5 min)
and resuspended in incomplete media and merozoites purified by
3 μm filtration (Whatman) immediately followed by 2 μm filtration
(Whatman).

## Inducing the conditional LoxPact1 cKO

The inducible act1 cKO was obtained by the addition of 50 nM rapa-
mycin to the parental LoxPAct1 strain for 4 h at 37°C, 5% $CO_2$ and
cultured as described previously [9].

## Inducing the conditional Cas9act1 cKO

The act1 cKO parasite strain was obtained by addition of 50 nM of
rapamycin to the parental lines. The strain was incubated for 1 h at
37°C and 5% $CO_2$ and cultured as described previously [9]. To
decrease the population of un-induced parasites, the culture media
was replaced by DMEM complete supplemented with 2.5% dextran
sulphate after 24 h to inhibit re-invasion of WT parasites.

## MyoA-SNAP staining

Parasites expressing MyoA-SNAP were incubated with a TMR ligand
for the SNAP tag [48] (1:10,000) for 30 min. The cells were then
washed three times with PBS prior analysis.

## SiR-tubulin staining

Prior invasion, parasites were incubated with 1,000 nM SiR-tubulin
[41] (1:1,000) for 1 h. The cells were then washed three times with
PBS prior analysis.

## Light microscopy

Widefield images were acquired in z-stacks of 2 μm increments and
were collected using an Olympus UPLSAPO 100× oil (1.40NA)
objective on a Delta Vision Core microscope (Applied Precision, GE)
attached to a CoolSNAP HQ2 CCD camera. Deconvolution was
performed using SoftWoRx Suite 2.0 (Applied Precision, GE). Video
microscopy was conducted with the Delta Vision Core microscope
as above. Normal growth conditions were maintained throughout
the experiment (37°C; 5% $CO_2$). Further image processing was
performed using FIJI [49] and Icy Image Processing Software (Insti-
tut Pasteur) [50].

Super-resolution microscopy (SR-SIM) was carried out using
an ELYRA PS.1 microscope (Zeiss) as described previously [24].
3D rendering and model views were generated using Imaris

software (Bitplane, Oxford Instruments) using the acquired SR-
SIM files.

## Flow analysis time-lapse microscopy

To discriminate parasite orientation, 500 nm of SiR-tubulin was
added to parasite cultures 1 h before the experiment. To prepare
tachyzoites for the assay, $1 \times 10^6$ mechanically freshly released
parasites per live dish were washed three times with PBS and incu-
bated a minimum of 10 min in serum-free DMEM media. Cytocha-
lasin D was added 30 min before imaging in a concentration of
2 μM. The samples were then imaged by widefield microscopy
using a Delta Vision Core microscope or super-resolution micro-
scopy (SR-SIM) using an ELYRA PS.1 microscope.

## Parasite treatment with A23187 and BIPPO

Calcium ionophore A23187 was added to parasite live dishes using
with a final concentration of 2μM. BIPPO was similarly used with a
final concentration of 5 μM. The parasites were imaged by light
microscopy immediately after adding A23187 or BIPPO.

## Tight Junction assay

For the assay $3 \times 10^6$, mechanically freshly released parasites per
well were incubated in 200 μl Endo buffer (44.7 mM $K_2SO_4$, 10 mM
$MgSO_4$, 106 mM sucrose, 5 mM glucose, 20 mM Tris–$H_2SO_4$,
3.5 mg/ml BSA, pH 8.2) for 10 min at 37°C. The parasites were then
allowed to settle into confluent layer of HFFs for 3 min at room
temperature. This step was followed by 20-min incubation at 37°C.
The supernatant was carefully removed and replaced by pre-
warmed culture media. The samples were further incubated for
5 min to allow invasion.

This was followed by fixing the samples using two different
buffers: cytoskeleton buffer (CB1) (MES pH 6.1 10 mM, KCL
138 mM, MgCl 3 mM, EGTA 2 mM, 5% PFA) and cytoskeleton
buffer (CB2) (MES pH6.1 10 mM, KCL 163.53 mM, MgCl
3.555 mM, EGTA 2.37 mM, sucrose 292 mM). These buffers were
mixed in a 4:1 ratio, respectively, and subsequently used for fixation
for 10 min. The samples were then treated with a PFA quenching
solution ($NH_4CL$ 50 mM) for 10 min followed by PBS washing for
three times. A minimum total of 40 parasites were counted for each
sample in three biological replicates. The tight junction assay
involving several mutants and wt in Fig 3E and F employed a multi-
ple comparison 2-way ANOVA using Tukey's multiple comparison
test with a $P$-value of < 0.0001.

## Cell deformation analysis

Samples were captured using SR-SIM and widefield microscopy as
previously described [24]. The deformation was then analysed with
Zen blue software (Zeiss) by measuring the size of the parasite body
and nucleus when going through a tight junction. A total of 100
parasites per biological replicate were measured for each condition
for RH in Fig 1B, and a total of 30 parasites per biological replicate
were measured for the mutant in Appendix Fig S1A. Statistical anal-
ysis was carried out by ordinary one-way ANOVA, using Tukey's
multiple comparison test with a $P$-value of < 0.0001.

## Colour-coded kymogram generation for particle dynamics analysis

Fourier-filtered colour-coded kymograms were generated using the KymographClear plugin on ImageJ and published in Mangeol *et al* [33]. This plugin allows to trace a path of interest on a collapse t-stack to follow particle movement. The plugin is able to generate a three colour-coded kymogram using Fourier transformation to filter different populations of particles by the orientation of the trajectory. The colour coding is represented by forward movement as red, backward movement as green and no movement as blue. These data are then exported to the stand-alone software KymographDirect to establish time-average intensity profiles. A more detailed protocol can be found on the author's website: https://sites.google.com/site/kymographanalysis/.

## Skeletonisation analysis

The skeletonisation process was done in FIJI using the Skeletonisation processing plugin [51]. The sample was binarised by thresholding. A skeletonisation algorithm is then employed which converts the thresholded signal into pixels, reducing the total width but not the length. The time-lapse profile was collapsed (Z-Project) to define the F-actin structures and their dynamics over the course of the time lapse. The plugin can be accessed by: https://imagej.net/Analyze Skeleton.

## Live cell invasion

Parasites were mechanically released using 23 G needle and filtered prior to inoculation on a confluent layer of HFFs, grown on glass bottom dishes (MaTek). The dish was then transferred to the DV Core microscope (Applied Precision, GE) and maintained under standard culturing conditions. Images were captured at 1.4 frames per second in DIC using an Olympus apochromat 60× oil objective. Images were analysed using the Icy Image Processing software (Institut Pasteur) [50]. Penetration speed profiles were obtained for 18 independent invasion events for RH Cb-EmeraldFP and Cb-EmeraldFP parasites. The statistical analysis of RH wt Invasion events in Fig 2C was done by unpaired two-tailed *t*-test in 28 total parasite invasion events that were captured across three different biological replicates.

## Penetration speed profile analysis

Movies obtained were analysed using Icy Image Processing software (Institut Pasteur) [50] with the Manual Tracking plugin. The methodology used consists on tracking the apical end of the parasite through the duration of penetration. The resulting data were then exported to GraphPad PRISM7 Software for posterior analysis.

## *Plasmodium* assays

The Cb-EmeraldFP sequence was cloned and transfected into *P. falciparum* 3D7 parasites, and parasites were cultured in RPMI 1640 supplemented with Albumax (Invitrogen). Schizonts were collected on a bed of 70% Percoll as previously described [42]. The schizonts were allowed to egress in a medium containing blood at 1% haematocrit and merozoites at various stages of invasion were fixed in 4% paraformaldehyde with 0.0075% glutaraldehyde as previously described [42]. The junction was stained with a RON4 antibody and the nucleus stained by DAPI. Cb-EmeraldFP-labelled actin filaments were fluorescent in the green range. For image acquisition, z-stacks were collected using a UPLSAPO 100× oil (1.40 NA) objective on a Delta Vision Core microscope (Applied Precision, GE) attached to a CoolSNAP HQ2 CCD camera. Deconvolution was performed using SoftWoRx Suite 2.0 (Applied Precision, GE). An Elyra S1 microscope with super-resolution structured illumination microscopy (SR-SIM) (Zeiss) was used for super-resolution dissection of RON4 staining on the merozoite surface.

*Plasmodium knowlesi* A1-H.1 parasites were cultured as described previously [47] in human O+ RBCs in RPMI-HEPES media supplemented with 2.3 g/l sodium bicarbonate, 2 g/l dextrose, 0.05 g/l hypoxanthine, 0.025 g/l gentamicin, 0.292 g/l L-glutamine, 5 g/l Albumax II (Gibco) and 10% (v/v) equine serum (Gibco). Parasites were cultured at 37°C with a gas mixture of 90% $N_2$, 5% $O_2$ and 5% $CO_2$.

PkRON2*mCherry-HA A1-H.1 merozoites were incubated with fresh erythrocytes, shaken at 1,000 rpm, 37°C for 90 s and samples fixed by addition of an equal volume of 2× fixative (8% paraformaldehyde/0.015% glutaraldehyde, Sigma-Aldrich). Cells were permeabilised with 0.1% Triton X-100/PBS for 10 min and blocked with 3% BSA/PBS. Cells were incubated with 1:500 mouse anti-actin [47] and 1:1,000 rat anti-HA (3F10, Roche) followed by 1:1,000 anti-mouse Alexa 488 (Invitrogen) and 1:1,000 anti-rat Alexa 594 (Invitrogen) secondary antibodies. Cells were smeared on slides and mounted in VectaShield (Vector Laboratories) with 0.1 ng/ml DAPI to label the parasite nucleus. Widefield fluorescent microscopy images were acquired with a Nikon plan apo 100×/1.45 oil immersion lens on a Nikon eclipse Ti microscope and images processed with NIS Elements (Nikon).

# Data availability

All imaging data and genetic material used for this paper are available from the authors on reasonable request.

Expanded View for this article is available online.

## Acknowledgments

We acknowledge the assistance of the Institute of Infection, Immunity and Inflammation Imaging Platform at the University of Glasgow. We also want to thank Dr. Philip Thompson for providing us with the reagent BIPPO and Dr. Maryse Lebrun for anti-RON2 antibodies. This work was supported by an ERC-Starting grant (ERC-2012-StG 309255-EndoTox), a Wellcome 087582/Z/08/Z Senior Fellowship to MM, Wellcome 100993/Z/13/Z Investigator Award to JB and a National Secretariat for Higher Education, Sciences, Technology and Innovation of Ecuador (SENESCYT) PhD scholarship (IFTH-GBE-2015-0475-M) to MD. The Wellcome Centre for Molecular Parasitology is supported by core funding from the Wellcome (085349). The funders had no role in study design, data collection and analysis, decision to publish, or preparation of the manuscript.

## Author contributions

MDR, JB, IT and MM conceived, designed or planned the study. MDR, JP, JAW, GP, OL, SD, GSP, JFS, LL and FL-B contributed to the acquisition, analysis or interpretation of the data. MM drafted the manuscript. All authors critically

reviewed or revised the manuscript for intellectual content and approved the final version.

## Conflict of interest

The authors declare that they have no conflict of interest.

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
