## [Review Process File · EMBO Reports]

Apicomplexan F-actin is required for efficient nuclear entry during host cell invasion

Mario Del Rosario, Javier Periz, Georgios Pavlou, Oliver Lyth, Fernanda Latorre-Barragan, Sujaan Das, Gurman S. Pall, Johannes Felix Stortz, Leandro Lemgruber, Jamie A. Whitelaw, Jake Baum, Isabelle Tardieux and Markus Meissner

Review timeline:	Submission date:	21st Jul 2019
	Editorial Decision:	15th Aug 2019
	Revision received:	21st Aug 2019
	Accepted:	11th Sep 2019

Editor: Deniz Senyilmaz Tiebe / Martina Rembold

Transaction Report:

(Note: This manuscript was transferred to *EMBO reports* following review at *The EMBO Journal*. With the exception of the correction of typographical or spelling errors that could be a source of ambiguity, letters and reports are not edited. The original formatting of letters and referee reports may not be reflected in this compilation.)

1st Editorial Decision

15th Aug 2019

Thank you for submitting your manuscript entitled 'Apicomplexan F-actin is required for efficient nuclear entry during host cell invasion' to EMBO Reports. Your manuscript was previously reviewed at another journal and you submitted a revised manuscript according to reviewers' comments. The original referees were not available anymore; therefore I contacted a third referee/arbitrating advisor, whose comments you can find below.

As you can see, the referee finds that you sufficiently addressed the concerns of the referees and recommends publication in this journal. Before I can accept the manuscript, I need you to address the below minor/editorial issues:

REFeree REPORTS:

Referee #1:

This is a revision that was originally reviewed for EMBO J by 2 reviewers and now is considered for EMBO Reports. The paper thoroughly and carefully analyses filamentous actin (visualised using a chromobody or antibodies) in invasion of *Toxoplasma* parasites into mammalian cells with some additional comparative work in *Plasmodium* parasites. It cleverly employs different mutants with limited invasion capacity to discern the distribution of F actin during invasion. The authors find a correlation between the presence of an actin ring that follows the constriction during invasion with the proportion of parasites depending on actin for invasion. Importantly, the authors note that the nucleus appears to be a major obstacle for invasion when it has to pass through the constriction. The authors find a pool of actin that is proximal to the nucleus and they build a compelling case that this F actin is involved in solving this problem for the parasite.

Overall this paper contains a lot of data that is of very high quality. The authors nicely visualise all the markers needed for their analysis and shed light on the invasion process using time lapse and superresolution imaging as well as IFAs. The imaging is impressive and phenotypes are all quantified. I do see the first reviewer's point that it is difficult to draw a definitive conclusion on the

role of actin in forcing passage of the nucleus but the data is very congruent with such a conclusion. In my opinion the authors do not overstate their results as presented in the revision and the concerns of this reviewer have therefore been solved. Reviewer 2 was interested in data on further Myosins but I fully agree with the authors that MyoA is the logic choice here. This is well explained in the current version of the paper. Even if other Myosins would contribute additional tools to this invasion analysis, the knock outs currently used are sufficient for the conclusions reached. Adding other Myosin knock outs would serve no further purpose for this study that is already quite dense in data. Overall the authors satisfactorily addressed the previous reviewer concerns and I find this work very suitable for EMBO Reports.

Should the opportunity present for a few last minor changes I would have the following questions and comments the authors might want to consider (page numbers are according to how the manuscript appears in the PDF viewer as no page numbers were included):

1. Introduction: We previously identified two F-actin polymerisation centres... ..In *T.gondii*, these three polymerisation centres of F-actin -> somewhat confusing, two or three?
2. page 11: 'and the PM is coordinated' -> add comma 'and the PM, is coordinated' or rephrase, as otherwise this is difficult to understand.
3. page 5: nucleus squeezing during invasion 'in a F-actin dynamic manner' -> it is not clear what this exactly means. Involving F actin? Movement? Force? Please specify.
4. The first results section, likely added in response to the previous reviewer 2, could also be part of the introduction if written slightly differently. This is by no means mandatory, but might be a better fit (it contains no data generated in this paper). This also applies to the first part of the second results section.
5. Page 6: 'Since transient transfections is typically associated with overexpression of the target protein hence with significant uncontrolled impact on a dynamic equilibrium [24], we generated *T. gondii* lines stably expressing Cb-EmeraldFP at comparable levels and...' please clarify to what the levels were comparable? To actin expression levels? Or was it meant that all parasites express comparable levels of the chromobody? Also: either 'transfections are' or change to the singular 'transfection'.
6. On what is the sekeletonisation of the signal in Figure 1B exactly based? I do not doubt the conclusions reached, but if the skeletons are based solely on the image shown in Figure 1A, I am not in all instances convinced this gives a meaningful representation of the original image. This also applies to Figure 1D and E where for instance the treated parasites in the top row do not seem to have much filamentous actin left in the parasite body but there are ample and very defined skeletons in that area. In all three cell lines the 0:31 time points with the arrows in Figure 1B are the ones with the highest intensity, is this a coincidence or is this a different representation of the data?
7. Page 7: Together, these data point that -> rephrase
8. Figure 1F: the labelling of the graph is somewhat cryptic. Was there also Jasplakinolide treatment? There are bars that are labelled 'jas'.
9. Page 9: 'the junctional ring appears remarkable constant' -> remarkably constant
10. Figure 2C has very small font and in general is very small, can this be enlarged?
11. In the ~20 of cell where there was no detectable actin ring during invasion, could this also be a detection problem because of less F actin in the area? In some of the images/movies it looks like there might be some fainter actin accumulations. Maybe it is just much less?
12. Model Figure 6 C? Did the schematic model fully capture what is seen in the microscopy images? At least in the images shown, the actin in (c) still seems in a circle even though it does not exactly co-locate with the Ron signal. In the model this looks quite different. Is this simply because in this cell F actin coincidentally forms a ring? In the last panel of the model (d) actin is missing. Again, it very much looks like it is in the same region as Ron in the corresponding microscopy image, in fact this time also co-localising. Or is this coincidental in the chose images?
13. Ron2 or Ron4, in the microscopy it is designated as Ron4, in the scheme as Ron2.
13. Legend Figure 6C: Model is on the right, microscopy on the left not the other way around, please correct.

Point-to-Point Reply to the reviewers

We would like to thank the reviewers for their efforts and valuable comments that have significantly contributed to improve the quality of our study.

Please find our reply in **blue**:

Referee #1:

This is a revision that was originally reviewed for EMBO J by 2 reviewers and now is considered for EMBO Reports. The paper thoroughly and carefully analyses filamentous actin (visualised using a chromobody or antibodies) in invasion of Toxoplasma parasites into mammalian cells with some additional comparative work in Plasmodium parasites. It cleverly employs different mutants with limited invasion capacity to discern the distribution of F actin during invasion. The authors find a correlation between the presence of an actin ring that follows the constriction during invasion with the proportion of parasites depending on actin for invasion. Importantly, the authors note that the nucleus appears to be a major obstacle for invasion when it has to pass through the constriction. The authors find a pool of actin that is proximal to the nucleus and they build a compelling case that this F actin is involved in solving this problem for the parasite.

Overall this paper contains a lot of data that is of very high quality. The authors nicely visualise all the markers needed for their analysis and shed light on the invasion process using time lapse and superresolution imaging as well as IFAs. The imaging is impressive and phenotypes are all quantified. I do see the first reviewer's point that it is difficult to draw a definitive conclusion on the role of actin in forcing passage of the nucleus but the data is very congruent with such a conclusion. In my opinion the authors do not overstate their results as presented in the revision and the concerns of this reviewer have therefore been solved. Reviewer 2 was interested in data on further Myosins but I fully agree with the authors that MyoA is the logic choice here. This is well explained in the current version of the paper. Even if other Myosins would contribute additional tools to this invasion analysis, the knock outs currently used are sufficient for the conclusions reached. Adding other Myosin knock outs would serve no further purpose for this study that is already quite dense in data. Overall the authors satisfactorily addressed the previous reviewer concerns and I find this work very suitable for EMBO Reports.

Should the opportunity present for a few last minor changes I would have the following questions and comments the authors might want to consider (page numbers are according to how the manuscript appears in the PDF viewer as no page numbers were included):

1. Introduction: We previously identified two F-actin polymerisation centres... ..In T.gondii, these three polymerisation centres of F-actin -> somewhat confusing, two or three?

We initially proposed the presence of two F-actin polymerisation centres based on our live microscopy observations on replicating parasites (see Periz *et al.*, 2017). After further work from our lab (Felix Stortz *et al.*, 2019) and from colleagues (Tosetti *et al.*, 2019), a third F-actin polymerisation centre was pointed in the residual body based on the location of formin-3.

2. page 11: 'and the PM is coordinated' -> add comma 'and the PM, is coordinated' or rephrase, as otherwise this is difficult to understand.

This paragraph is now corrected.

3. page 5: nucleus squeezing during invasion 'in a F-actin dynamic manner' -> it is not clear what this exactly means. Involving F actin? Movement? Force? Please specify.

This paragraph was slightly changed to be more specific.

4. The first results section, likely added in response to the previous reviewer 2, could also be part of the introduction if written slightly differently. This is by no means mandatory, but might be a better fit (it contains no data generated in this paper). This also applies to the first part of the second results section.

We discussed this possibility. However, we found it is most fitting in the result section, since it explains our rationale and would stand quite isolated in the introduction. As the reviewer rightly mentions, we added this section in response to previous reviewer 2 to make our rationale more clear to a broader readership.

5. Page 6: 'Since transient transfections is typically associated with overexpression of the target protein hence with significant uncontrolled impact on a dynamic equilibrium [24], we generated T. gondii lines stably expressing Cb-EmeraldFP at comparable levels and...' please clarify to what the levels were comparable? To actin expression levels? Or was it meant that all parasites express comparable levels of the chromobody? Also: either 'transfections are' or change to the singular 'transfection'.

This is now corrected.

6. On what is the skeletonisation of the signal in Figure 1B exactly based? I do not doubt the conclusions reached, but if the skeletons are based solely on the image shown in Figure 1A, I am not in all instances convinced this gives a meaningful representation of the original image.

In all skeletonisation instances, the images depicted in the panels 1B,D,E are based on maximum projections of the timepoints indicated in the figure and shown in supplementary movies. We chose to present the images in this way since the skeletonisation process detects well-defined signals in each frame, resulting in few "skeletons" per individual frame. By projecting the images over time, a complete structures can be observed that depicts the dynamics and localisation of F-actin in a more informative way. The figure legend was changed to correctly inform these panels are projections of their respective timepoints.

This also applies to Figure 1D and E where for instance the treated parasites in the top row do not seem to have much filamentous actin left in the parasite body but there are ample and very defined skeletons in that area.

The increase in "skeletons" in Figure 1E is due to how the skeletonisation plugin works. In order to convert the signal of our F-actin into pixels or "skeletons" an automatic threshold is applied into the image, converting the initial signal (with many different intensities) into a binary signal (either white or black). This automatic threshold process converted some of the weaker signals into skeletons as well, resulting in the image we presented in panel 1E.

In all three cell lines the 0:31 time points with the arrows in Figure 1B are the ones with the highest intensity, is this a coincidence or is this a different representation of the data?

As mentioned above, the time point 0:31 is the cumulative projection of 31 seconds. The increase in intensity resulted in the accumulation of more "skeletons" across the movie.

7. Page 7: Together, these data point that -> rephrase

The sentence was rephrased.

8. Figure 1F: the labelling of the graph is somewhat cryptic. Was there also Jasplakinolide treatment? There are bars that are labelled 'jas'.

We apologise for this oversight. We initially used JAS in addition to CD treatments. However, due to the stabilisation of F-actin by Jas, no evident effect upon A23187 or BIPPO treatment can be observed.

9. Page 9: 'the junctional ring appears remarkable constant' -> remarkably constant
This is now corrected.

10. Figure 2C has very small font and in general is very small, can this be enlarged?
This is now corrected.

11. In the ~20 of cell where there was no detectable actin ring during invasion, could this also be a detection problem because of less F actin in the area? In some of the images/movies it looks like there might be some fainter actin accumulations. Maybe it is just much less?

We agree with the reviewer that in WT parasites different levels of accumulation can be observed and that in some instances, even though F-actin is not detectable, it cannot be ruled out that no F-actin is present. The situation is different for the mutants, where no F-actin accumulation could be observed at the junction. We speculate that there is a fine tuning of the amount of F-actin required at the junction to facilitate invasion.

12. Model Figure 6 C? Did the schematic model fully capture what is seen in the microscopy images? At least in the images shown, the actin in (c) still seems in a circle even though it does not exactly co-localise with the Ron signal. In the model this looks quite different. Is this simply because in this cell F actin coincidentally forms a ring?

The model was updated. The selected images were representatives and the model slightly oversimplified.

In the last panel of the model (d) actin is missing. Again, it very much looks like it is in the same region as Ron in the corresponding microscopy image, in fact this time also co-localising. Or is this coincidental in the chose images?

See above

13. Ron2 or Ron4, in the microscopy it is designated as Ron4, in the scheme as Ron2.

For *T. gondii* work, anti-RON2 antibodies were used. For *P. falciparum* anti-RON4 antibodies were employed and in the case of *P. knowlesi*, pkRON2 was tagged using mCherry.

13. Legend Figure 6C: Model is on the right, microscopy on the left not the other way around, please correct.

This is now corrected.

2nd Editorial Decision

11th Sep 2019

Thank you for sending your revised manuscript. Since Deniz is currently away from the office, I have stepped in as the secondary editor of this manuscript. I have now looked at everything and all looks fine. I am therefore very pleased to accept your manuscript for publication in EMBO Reports.

Congratulations on a very nice work!

Corresponding Author Name: Markus Meissner

Manuscript Number: EMBOR-2019-48896V2